# ASSESSING VISUALLY-CONTINUOUS CORRUPTION RO-BUSTNESS OF NEURAL NETWORKS RELATIVE TO HU-MAN PERFORMANCE

## ABSTRACT

While Neural Networks (NNs) have surpassed human accuracy in image classification on ImageNet, they often lack robustness against image corruption, i.e., corruption robustness. Yet such robustness is seemingly effortless for human perception. In this paper, we propose *visually-continuous corruption robustness* (VCR) – an extension of corruption robustness to allow assessing it over the wide and continuous range of changes that correspond to the human perceptive quality (i.e., from the original image to the full distortion of all perceived visual information), along with two novel human-aware metrics for NN evaluation. To compare VCR of NNs with human perception, we conducted extensive experiments on 14 commonly used image corruptions with 7,718 human participants and state-of-the-art robust NN models with different training objectives (e.g., standard, adversarial, corruption robustness), different architectures (e.g., convolution NNs, vision transformers), and different amounts of training data augmentation.

Our study showed that: 1) assessing robustness against continuous corruption can reveal insufficient robustness undetected by existing benchmarks; as a result, 2) the gap between NN and human robustness is larger than previously known; and finally, 3) some image corruptions have a similar impact on human perception, offering opportunities for more cost-effective robustness assessments. Our validation set with 14 image corruptions, human robustness data, and the evaluation code is provided as a toolbox and a benchmark.

## 1 INTRODUCTION

For Neural Networks (NN), achieving robustness against possible corruption (i.e., corruption robustness) that can be encountered during deployment is essential for the application of NN models in safety-critical domains (Hendrycks & Dietterich, 2019). Since NN models in these domains automate tasks typically performed by humans, it is necessary to compare the model's robustness with that of humans.

**Human versus NN robustness.** Corruption robustness measures the average-case performance of an NN or humans on a set of image corruption functions (Hendrycks & Dietterich, 2019). Existing studies, including out-of-distribution anomalies (Hendrycks & Gimpel, 2017), benchmarking (Hendrycks & Dietterich, 2019; Hendrycks et al., 2021b), and comparison with humans (Hu et al., 2022; Geirhos et al., 2021), generally evaluate robustness against a pre-selected, fixed set of transformation parameter values that represent varying degrees of image corruption. However, parameter values cannot accurately represent the degree to which human perception is affected by image corruptions. For instance, using the same parameter to brighten an already bright image will make the objects harder to see but will have the opposite effect on a dark image (Hu et al., 2022). Additionally, humans can perceive and generalize across a wide and continuous spectrum of visual corruptions from subtle to completely distorted (Geirhos et al., 2019a; Sheikh & Bovik, 2006). Relying solely on preset parameter values for test sets could lead to incomplete coverage of the full range of visual corruptions, leading to potential biases in evaluation results, which cannot accurately represent NN robustness compared with humans.

**Contributions and Outlook.** To address the above problem, we propose a new concept called *visually-continuous corruption robustness* (VCR). This concept focuses on the robustness of neural networks (NN) against a continuous range of image corruption levels. Additionally, we introduce two novel human-aware NN evaluation metrics (HMRI and MRSI) to assess NN robustness in comparison to human performance. We conducted extensive experiments with 7,718 human participants on the Mechanical Turk platform on 14 commonly used image transformations. Comparing NN and human VCR with our metrics, we found that a significant robustness gap between NNs and humans still exists: no model can fully match human performance throughout the entire continuous range in terms of both accuracy and prediction consistency, and few models can exceed humans by a small margin in specific levels of corruption. Furthermore, our experiments yield insightful findings about robustness of human and state-of-the-art (SoTA) NNs concerning accuracy, degrees of visual corruption, and consistency of classification, which can contribute towards the development of NNs that match or surpass human perception. We also discovered classes of corruption transformations for which humans showed similar robustness (e.g., different types of noise), while NNs reacted differently. Recognizing these classes can contribute to reducing the cost of measuring human robustness and elucidating the differences between humans and computational models. To foster future research, we open-sourced all human data as a comprehensive benchmark along with a Python code that enables test set generation, testing, and retraining.

## 2 METHODS: VCR, TESTING, METRICS, CROWDSOURCING, NN MODELS

To study NN robustness against a wide and continuous spectrum of visual changes, we first define the VCR and then describe our method for generating test sets. To study VCR of NNs in relation to humans, we also present the human-aware metrics, followed by human robustness data and NN models used in the study.

**Visually-Continuous Corruption Robustness (VCR).** A key difference between corruption robustness and VCR is that the latter is defined relative to the *visual impact* of image corruption on human perception, rather than the transformation parameter domain. To quantify visual corruption, VCR uses the Image Quality Assessment (IQA) metric Visual Information Fidelity (*VIF*) (Sheikh & Bovik, 2006; Kumar, 2020). VIF measures the perceived quality of a corrupted image $x'$ compared to its original form $x$ by measuring the visual information unaffected by the corruption. Thus, we define the *change* in the perceived quality caused by the corruption as $\Delta_v(x, x') = max(0, 1 - VIF(x, x'))$. See the appendix for more detail on $\Delta_v$. With $\Delta_v$, whose value ranges from 0 and 1, we can consider VCR against the wide, finite, and continuous spectrum of visual corruptions ranging from no degradation to visual quality (i.e., the original image) ($\Delta_v = 0$) to the full distortion of all visual information ($\Delta_v = 1$).

Limitation: VCR is limited to image corruption that is applicable to the chosen IQA metric, thus by using VIF, VCR is limited to only pixel-level corruption. Further research is needed for metrics suitable for other types of corruption (e.g., geometric).

For VCR, we consider a classifier NN $f : X \rightarrow Y$ trained on samples of a distribution of input images $P_X$, a ground-truth labeling function $f^*$, and a parameterized image corruption function $T_X$ with a parameter domain $C$. We wish to consider the robustness of $f$ against images with all degrees of visual corruption *uniformly* ranging from $\Delta_v = 0$ to $\Delta_v = 1$.[1] Therefore, given a value $v \in [0, 1]$, we define $P(x, x'|v)$ as the *joint distribution* of original images ($x$) and corresponding corrupted images ($x' = T_X(x, c), c \in C$) with $\Delta_v(x, x') = v$. VCR is defined in the presence of a robustness property $\gamma$ that $f$ should satisfy in the presence of $T_X$:

$$\mathcal{R}_\gamma = \mathbb{E}_{v \sim Uniform(0,1)}(P_{x,x' \sim P(x,x'|v)}(\gamma)). \tag{1}$$

In this paper, we instantiate VCR with two existing robustness properties (see Fig. 7 in the appendix). The first one is *accuracy (a)*, requiring that the prediction on corrupted images should be correct, i.e., $f(x') = f^*(x)$. It is also used in the existing definition of corruption robustness (Hendrycks & Dietterich, 2019). Thus,

$$\mathcal{R}_a = \mathbb{E}_{v \sim Uniform(0,1)}(P_{x,x' \sim P(x,x'|v)}(f(x') = f^*(x))). \tag{2}$$

---

[1]Note that distributions other than uniform can be used based on the application. For example, one may wish to favour robustness against heavy snow conditions for NNs deployed in arctic areas.

The second property is *prediction consistency (p)*, requiring consistent predictions before and after corruptions, i.e., $f(x') = f(x)$ (Hu et al., 2022). It is applicable when ground truth is not available, which is common during deployment. Thus,

$$\mathcal{R}_p = \mathbb{E}_{v \sim Uniform(0,1)}(P_{x,x' \sim P(x,x'|v)}(f(x') = f(x))). \quad (3)$$

**Testing VCR.** VCR of a subject (a human or an NN) is measured by first generating a test set through sampling and then estimating it using the sampled data. The test set is generated by sampling images and applying corruption to obtain $P(x, x'|v)$ for different $\Delta_v$ values $v$. We sample $x \sim P_X$ and $c \sim Uniform(C)$, and obtain $x' = T_X(x, c)$ and $v = \Delta_v(x, x')$, resulting in samples $(x, x', c, v)$. Then, we divide them into groups of $(x, x', c)$, each with the same $v$ value. Next, by dropping $c$, we obtain groups of $(x, x')$ with the same $v$, which are samples from $P(x, x'|v)$. Note that this procedure requires only sufficient data in each group but not uniformity, i.e., $v \sim Uniform(0, 1)$ is not required. The varying size of each group, i.e., the non-uniformity of $v$ distribution, will not distort VCR estimates, but only impact the estimate uncertainty at a given $v$. Further, interpolation in the next step helps address any missing points (see Alg. 1 in the appendix).

With the test set, we estimate the performance w.r.t. the property $\gamma$ for each $v$. For each $v$ in the test data, we compute the *rate* of accurate predictions $f(x') = f^*(x)$ to estimate accuracy, i.e., $a_v = P_{x,x' \sim P(x,x'|v)}(f(x') = f^*(x))$ [resp. consistent predictions $f(x') = f(x)$ to estimate consistency, i.e., $p_v = P_{x,x' \sim P(x,x'|v)}(f(x') = f(x))$]. Then by plotting $(v, a_v)$ and $(v, p_v)$ and applying monotonic smoothing splines (Koenker et al., 1994) to reduce randomness and outliers, we obtain smoothed spline curves $s_a$ and $s_p$, respectively. The curves $s_\gamma$ (namely, $s_a$ and $s_p$) describe how the performance w.r.t. the robustness property $\gamma$ (namely, $a$ and $p$) decreases as the visual corruption in images increases. Finally, we estimate $\mathcal{R}_a = \mathbb{E}_{v \sim Uniform(0,1)}(a_v)$ [resp. $\mathcal{R}_p = \mathbb{E}_{v \sim Uniform(0,1)}(p_v)$] as the area under the spline curve, i.e., $\hat{\mathcal{R}}_a = A_a = \int_0^1 s_a(v)dv$ [resp. $\hat{\mathcal{R}}_p = A_p = \int_0^1 s_p(v)dv$].

**Human-Aware Metrics.** A commonly used metric for measuring corruption robustness is the *Corruption Error (CE)* (Hendrycks & Dietterich, 2019)—the top-1 classification error rate on the corrupted images, normalized by the error rate of a baseline model. CE can be used to compare an NN with humans if the baseline model is set to be humans. However, CE is not able to determine whether an NN can exceed humans, and NN models could potentially have super-human accuracy for particular types of perturbations or in some $\Delta_v$ ranges. Therefore, inspired by CE, we propose two new human-aware metrics, *Human-Relative Model Robustness Index (HMRI)* that measures NN VCR relative to human VCR; and *Model Robustness Superiority Index (MRSI)* that measures how much an NN exceeds human VCR. These metrics take both the estimated spline curve for humans, $s_\gamma^h$, and for NN, $s_\gamma^m$, as inputs, and we denote areas under these curves as $A_\gamma^h$ and $A_\gamma^m$, respectively (see Fig. 8).

**Definition 1 [Human-Relative Model Robustness Index (HMRI)].** Given $s_\gamma^h$ and $s_\gamma^m$, let $A_\gamma^{h>m} = \int_0^1 (s_\gamma^h(v) - s_\gamma^m(v))^+ dv$ denote the average (accuracy or preservation) performance lead of humans over a model across the visual change range, where the performance lead is defined as the positive part of performance difference, i.e., $(s_\gamma^h(v) - s_\gamma^m(v))^+ = max(0, s_\gamma^h(v) - s_\gamma^m(v))$. *HMRI*, which quantifies the extent to which a DNN can replicate human performance, is defined as $\frac{A_\gamma^h - A_\gamma^{h>m}}{A_\gamma^h} = 1 - \frac{A_\gamma^{h>m}}{A_\gamma^h}$.

The *HMRI* value ranges from $[0, 1]$; a higher *HMRI* indicates a NN model closer to human VCR, and *HMRI* = 1 signifies that $s^m$ is the same as or completely above $s^h$ in the entire $\Delta_v$ domain, meaning that the NN is at least as reliable as a human (see Fig. 8 in the appendix).

**Definition 2 [Model Robustness Superiority Index (MRSI)].** Given $s_\gamma^h$ and $s_\gamma^m$, let $A_\gamma^{m>h} = \int_0^1 (s_\gamma^m(v) - s_\gamma^h(v))^+ dv$ denote the average performance lead of a model over a human across the visual change range. *MRSI*, which quantifies the extent to which a DNN model can surpass human performance, is defined as $\frac{A_\gamma^{m>h}}{A_\gamma^m}$.

The *MSRI* value ranges from $[0, 1)$, with the higher value indicating better performance than humans. *MSRI* = 0 means that the given NN model performs worse than or equal to humans in the entire $\Delta_v$ domain. A positive *MSRI* value indicates that the given NN model performs better than humans at least in some ranges of $\Delta_v$ (see Fig. 8).

Comparing humans and NNs with *HMRI* and *MRSI* yields three possible scenarios: (1) humans' performance fully exceeds NN's, i.e., $0 < HMRI < 1$ and $MRSI = 0$; (2) NN's performance fully

exceeds humans', i.e., *HMRI* = 1 and *MRSI* > 0 ; and (3) humans' performance is better than NN's in some $\Delta_v$ intervals and worse in others, i.e., *HMRI* < 1 and *MRSI* > 0.

**Image Corruptions.** In this paper, we focus on studying VCR of NNs in relation to humans regarding 14 commonly used image corruptions from three different sources: Shot Noise, Impulse Noise, Gaussian Noise, Glass Blur, Gaussian Blur, Defocus Blur, Motion Blur, Brightness and Frost from IMAGENET-C (Hendrycks & Dietterich, 2019); Blur, Median Blur, Hue Saturation Value and Color Jitter from Albumentations (Buslaev et al., 2020); and Uniform Noise from Geirhos et al. (2019a). See the appendix for a visualization of these corruptions.

**Crowdsourcing.** Given that VCR is focused on the average-case performance, we chose to use crowdsourcing for measuring human performance. This allowed us to involve a large number of participants for a more precise estimation of the average-case human performance. The experiment is designed following (Hu et al., 2022) and (Geirhos et al., 2019a). The experiment procedure is a *forced-choice image categorization task*: humans are presented with one image at a time, for 200 ms to limit the influence of recurrent processing, and asked to choose a correct category out of 16 entry-level class labels (Geirhos et al., 2019a). For NN models, the 1,000-class decision vector was mapped to the same 16 classes using the WordNet hierarchy (Geirhos et al., 2019a). The time to classify each image was set to ensure fairness in the comparison between humans and machines (Firestone, 2020). Between images, we showed a noise mask to minimize feedback influence in the brain (Geirhos et al., 2019a). We included qualification tests and sanity checks aimed to filter out cases of participants misunderstanding the task and spammers (Papadopoulos et al., 2017), and only considered results from those participants who passed both tests. As a result, we had 7,718 participants and obtained approximately (1) 70,000 human predictions on images with different levels of visual corruptions; and (2) 50,000 human predictions on original images as these can be repeated in experiments for different corruptions. The same original image, corrupted or not, was never shown again to the same participant.

**NN models.** We studied 11 standard supervised models: NOISYMIX, NOISYMIX_NEW (Erichson et al., 2022), SIN, SIN_IN, SIN_IN_IN, HMANY, HAUGMIX (Hendrycks et al., 2020), STANDARD_R50 (Paszke et al., 2019), ALEXNET (Krizhevsky et al., 2012); 4 adversarial learning models: DO_50_2_LINF (Salman et al., 2020), LIU-SWIN-L, LIU-CONVNEXT-L (Liu et al., 2023), SINGH-CONVNEXT-L-CONVSTEM (Singh et al., 2023); 2 SWSL models: swsl_resnet18, swsl_resnext101_32x16D (Yalniz et al., 2019); 3 ViT models: TIAN_DEIT-S, TIAN_DEIT-B (Tian et al., 2022), DINOV2_GIANT (Oquab et al., 2023); and 1 CLIP (clip-vit-base-patch32) model (Radford et al., 2021). For CLIP, we used a simple prompt "a picture of (ImageNet class)" while tokenizing the labels. See the appendix for more details on the models and their selection.

## 3 TESTING ROBUSTNESS AGAINST VISUAL CORRUPTION

IMAGENET-C is the SoTA benchmark for corruption robustness. Rather than considering the continuous range of corruption like VCR, IMAGENET-C includes all IMAGENET validation images corrupted using 5 pre-selected parameter values for each type of corruption (Hendrycks & Dietterich, 2019). This section compares robustness measured with IMAGENET-C vs. VCR on all 9 IMAGENET-C corruption functions in our study. Due to the page limit, we include full results in the appendix.

**Visual Corruption in Test Sets.** For each corruption, our tests generated for checking VCR contain $50,000$ images, mirroring the size of the IMAGENET (Russakovsky et al., 2015) validation set, while IMAGENET-C includes $5 \times 50,000$ images. Because of the difference in how test sets are generated, we can observe two major differences in the distributions of degrees of visual corruption: they have different coverage and peak at different values (e.g., Fig. 1).

To quantitatively assess the actual coverage of $\Delta_v$ in the test sets, Tab. 1 gives the coverage as a percentage of the full $\Delta_v$ range of $[0, 1]$. To compute it, the distribution is divided into 40 bins with the same width. A bin is considered covered if it contains 20 or more images. The coverage is then determined by dividing the number of covered bins by the total number of bins (40). We observed that IMAGENET-C exhibits a low coverage of $\Delta_v$ values. Specifically, as shown in Fig. 1 and Tab. 1, the distribution of IMAGENET-C in Gaussian blur has coverage of only 56.4% focusing mainly on the center of the entire domain of $\Delta_v$ and missing coverage for low and high $\Delta_v$ values, which can lead to biased evaluation. As we show in the appendix, the same can be observed for most

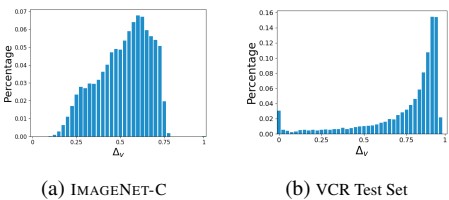

(a) IMAGENET-C

(b) VCR Test Set

Figure 1: Histograms showing $\Delta_v$ distribution between IMAGENET-C and our VCR test sets for Gaussian Blur.

Table 1: $\Delta_v$ Coverage Comparison with IMAGENET-C.

| Corruption | Coverage | |
|---|---|---|
| | IMAGENET-C | VCR Test Set |
| Brightness | 0.590 | 1.000 |
| Gaussian Blur | 0.564 | 0.974 |
| Defocus Blur | 0.538 | 0.923 |
| Shot Noise | 0.462 | 0.590 |
| Frost | 0.436 | 1.000 |
| Gaussian Noise | 0.436 | 0.872 |
| Impulse Noise | 0.385 | 0.641 |
| Motion Blur | 0.333 | 0.974 |
| Glass Blur | 0.333 | 0.949 |

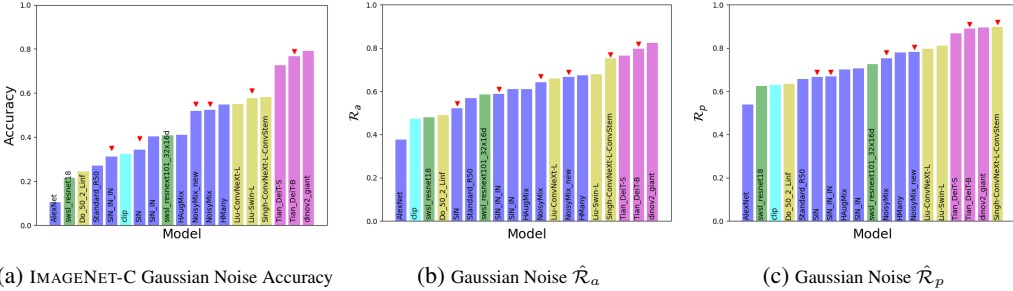

(a) IMAGENET-C Gaussian Noise Accuracy    (b) Gaussian Noise $\hat{\mathcal{R}}_a$    (c) Gaussian Noise $\hat{\mathcal{R}}_p$

Figure 2: Comparison between IMAGENET-C and VCR with Gaussian Noise. Models discussed in the text are marked by a red triangle.

IMAGENET-C corruption functions. On the other hand, our test sets provide coverage for almost the entire domain, with a coverage percentage of 97.4%. This pattern holds true for other corruption functions as well—our test sets have consistently higher coverage than IMAGENET-C.

As for VCR, Shot Noise and Impulse Noise have relatively low coverage, because the level of noise these functions add is exponential to their parameters. As a result, uniform sampling of the parameter range $C$ fails to cover small $\Delta_v$ values. When using uniform sampling over $C$, reaching the full coverage of $\Delta_v$ would require a large amount of data. Note, however, Alg. 1 still computes VCR over the full $\Delta_v$ range of $[0..1]$, and the lack of samples for low values of $\Delta_v$ has a limited impact on the VCR estimate. This is because we fit a monotonic spline that is anchored with a known initial performance for $\Delta_v = 0$, as discussed in the appendix.

Remark: The reported accuracy of IMAGENET-C can be directly impacted both by a lack of coverage and by non-uniformity, as it is computed as the average accuracy of all transformed images. In contrast, the shape of the $\Delta_v$ distribution in the generated test set does not impact VCR once sufficient coverage is achieved to estimate the spline curves $s_\gamma$, as already explained.

**Robustness Evaluation Results.** Next, we compare robustness evaluation results obtained with IMAGENET-C and VCR test sets. Consider results for Gaussian Noise in Fig. 2. NOISYMIX and NOISYMIX_NEW have almost the same robust accuracy on IMAGENET-C, but NOISYMIX_NEW has higher $\hat{\mathcal{R}}_a$; similarly, SIN has higher IMAGENET-C robust accuracy but lower $\hat{\mathcal{R}}_a$ than SIN_IN_IN. This is due to the almost complete lack of coverage for $\Delta_v < 0.5$ for Gaussian Noise in IMAGENET-C (see Tab. 1 and Fig. 11f), which can lead to biased evaluation results (i.e., biased towards $\Delta_v \geq 0.5$). Checking VCR allows us to detect such biases.

In addition to accuracy, VCR can also be used to check whether the NN can preserve its predictions after corruption, i.e., the prediction consistency property $p$, which can give us additional information about NN robustness. From Fig. 2b,c we can see that the model TIAN_DEIT_B has a higher $\hat{\mathcal{R}}_a$ than SINGH-CONVNEXT-L-CONVSTEM but a lower $\hat{\mathcal{R}}_p$. This suggests that even though TIAN_DEIT_B has better accuracy for corrupted images, it labels the same image with different labels before and after the corruption. Since ground truth would be hard to obtain during deployment, having low prediction consistency indicates issues with model stability and could raise concerns about when to trust the model prediction. The results for the remaining corruptions are in the appendix.

Summary: It is essential to test robustness before deploying NNs into an environment with a wide and continuous range of visual corruptions. Our results confirmed that testing robustness in this range

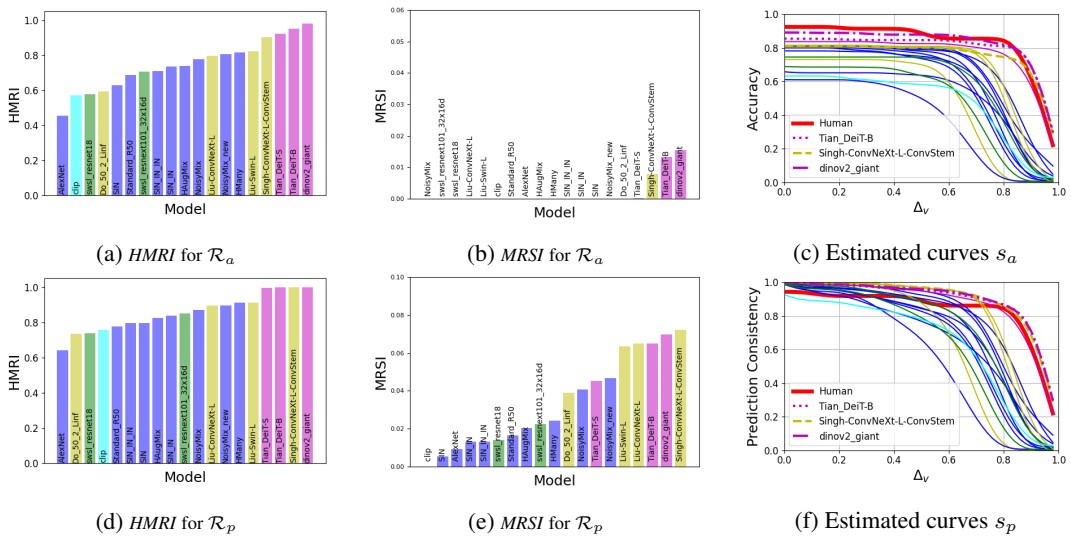

Figure 3: VCR evaluation results for Gaussian Noise. Results include, for each NN, the estimated curves $s_a$ and $s_p$ (representing how the performance w.r.t. the robustness properties $a$ and $p$ decreases as $\Delta_v$ increases); and the corresponding *HMRI* and *MRSI* values. Results are colored based on their category: Human, Vision Transformer, Supervised Learning, SWSL, Adversarial Training, CLIP.

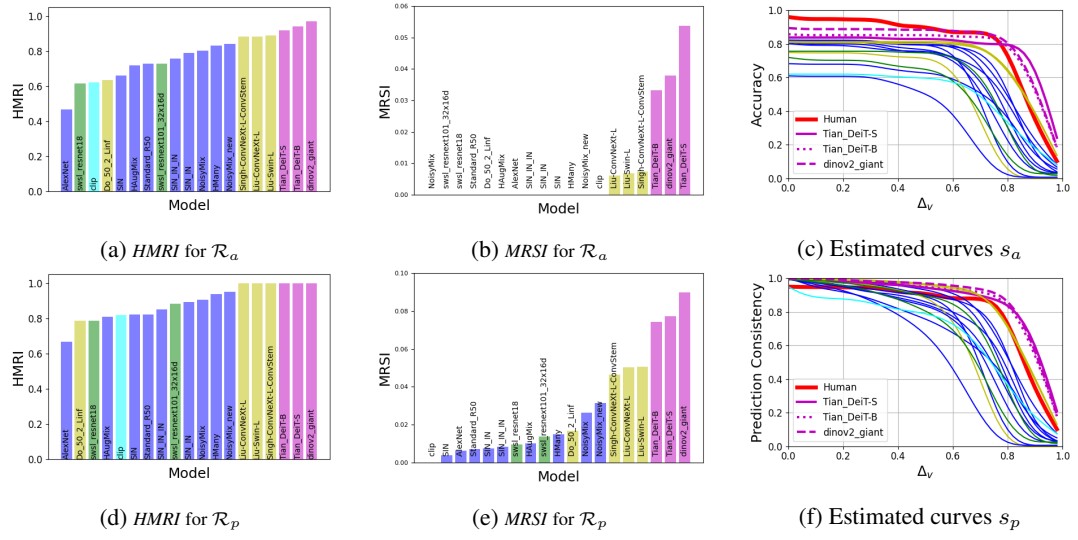

Figure 4: VCR evaluation results for Uniform Noise.

using a fixed and pre-selected number of parameter values can lead to undetected robustness issues, which can be avoided by checking VCR. Additionally, accuracy cannot accurately represent model stability when facing corruptions, which can be addressed by testing $\mathcal{R}_p$.

## 4 VCR OF DNNs COMPARED WITH HUMANS

We use our two new human-aware metrics, *HMRI* and *MRSI*, and the data from the human experiment to compare VCR of the studied models against human performance as a baseline.

For Gaussian Noise, Fig. 3 presents our measured *HMRI* and *MRSI* values for $\mathcal{R}_a$ and $\mathcal{R}_p$. For both metrics, a higher value indicates better robustness. As shown in Fig. 3a, no NN has reached 1.0 for *HMRI*$_a$, and in Fig. 3d, only 3 out of 21 NNs DINOV2_GIANT, TIAN_DEIT-B and SINGH-CONVNEXT-L-CONVSTEM reached 1.0 for *HMRI*$_p$, indicating that there are still unclosed gaps

between human and NN robustness, with humans giving more accurate and more consistent predictions facing corruptions than most SoTA NNs. These thee top-performing models have also the highest *HMRI* values for both $\mathcal{R}_a$ and $\mathcal{R}_p$, making these models closest to human robustness. In Fig. 3b, we can see that these three models have $MRSI_a$ values above 0.0, indicating that they surpass human accuracy in certain ranges of visual corruption. This can be visualized by checking the estimated curves $s_a$ as shown in Fig. 3c. The top-three models exceed human accuracy (the red curve) when $\Delta_v > 0.85$. For prediction consistency, Fig. 3e shows that all NNs have the $MRSI_p$ value above 0.0 and this is because, as shown in Fig. 3f, all NN curves are above the human curve when the $\Delta_v$ value is small. Specifically, the top-three models completely exceed humans in the entire $\Delta_v$ range.

Similarly, for Uniform Noise, as shown in Fig. 4a and Fig. 4d, no models reached 1.0 for $HMRI_a$ and the top-three models, reached 1.0 for $HMRI_p$. Together with Fig. 4b and Fig. 4e, we can see that for both $\mathcal{R}_a$ and $\mathcal{R}_p$, TIAN_DEIT-B has higher *HMRI* values but TIAN_DEIT-S has higher *MRSI* values. This suggests that while TIAN_DEIT-B is closer to human performance, TIAN_DEIT-S exceeds human performance more. This result may be counter-intuitive but can be explained with the curves $s_a$ and $s_p$ representing how the performance w.r.t. the robustness properties $a$ and $p$ decreases as $\Delta_v$ increases, as shown in Fig. 4c and Fig. 4f. From both $s_a$ and $s_p$, we observed that for $\Delta_v$ values less than 0.8, the performance of TIAN_DEIT-B is higher than TIAN_DEIT-S and closer to human, hence the higher *HMRI* value; and after $\Delta_v = 0.8$ when human performance starts decreasing, the TIAN_DEIT-B performance drops rapidly to much below that of TIAN_DEIT-S, hence the lower *MRSI* value.

This suggests that both *HMRI* and *MRSI* are useful for comparing NN robustness, and our curves $s_a$ and $s_p$ can provide further information on NN robustness with different degrees of visual corruption.

Overall, in both Fig. 3 and Fig. 4, we observed that the three ViT models (shown in purple) have the best performance for both $\mathcal{R}_a$ and $\mathcal{R}_p$, making them the models closest to human robustness. The same can also be observed for the rest of the corruption functions; see the appendix for more details. This indicates that vision transformer is the most promising architecture for reaching human-level robustness, even outperforming models trained with additional training data. The data in the appendix also indicates the biggest remaining robustness gap for blur corruptions. Furthermore, as we show in the appendix, our generated test sets can be used during model retraining for improved robustness compared to humans, resulting in with higher *HMRI* and *MRSI* values.

Summary: As our results suggest, when considering the full range of visually-continuous corruption, no NNs can match human accuracy, especially for blur corruptions, and only the best-performing ones can match human prediction consistency. For some specific degrees of corruption, few NNs can exceed humans by mostly tiny margins. This highlights a more substantial gap between human and NN robustness than previously identified by Geirhos et al. (2021). By evaluating VCR using our human-centric metrics, we gain deeper insights into the robustness gap, which can aid in the development of models closer to human robustness.

## 5 VCR FOR VISUALLY SIMILAR CORRUPTION FUNCTIONS

One noteworthy observation we made from our experiments with humans is the existence of *visually similar* corruption functions. This can contribute towards reducing experiment costs and a better understanding of differences between humans and NNs.

Different corruptions change different aspects of the images, e.g., image colour, contrast, and the amount of additive visual noise, and thus affect human perception differently (Geirhos et al., 2019a). Also, multiple different corruption functions can be implemented for the same visual effect, such as Gaussian noise and Impulse noise for noise addition. Although the difference between Gaussian noise and Impulse noise can be picked up by complex NN models, an average human would struggle to distinguish between the two. Therefore, for a specific visual effect, there should exist a class of corruption functions implementing the effect that an average human is unable to tell them apart. We call corruption functions in the same class *visually similar*. We postulate that since visually similar functions, by definition, affect human perception similarly, they would affect human robustness similarly as well. Therefore, human data for one function can be reused for other similar functions in the same class possibly reducing experiment costs.

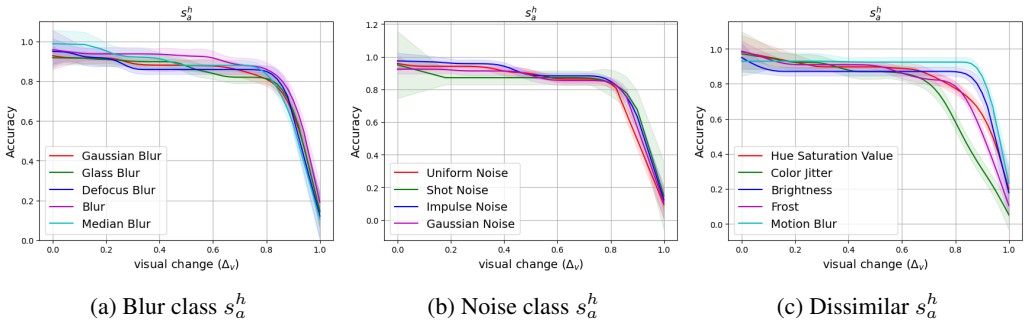

(a) Blur class $s_a^h$    (b) Noise class $s_a^h$    (c) Dissimilar $s_a^h$

Figure 5: Comparing human performance spline curves $s_a^h$ for similar and dissimilar corruption functions. For each curve, the coloured region around the curve is the $83\%$ confidence interval used for comparison of similarity. See $s_p^h$ in supplementary materials.

Since VCR is estimated with the spline curves $s_a^h$ and $s_p^h$, if the difference among the curves of a set of functions is statistically insignificant, human data (i.e., the spline curves) can be reused among the functions in this set. In Fig. 5, we plot the smoothed spline curves $s_a^h$ and $s_p^h$ obtained for all 14 corruption functions included in our experiments. We can observe that, for all corruption functions shown, human performance decreases slowly for small values of visual degrade ($\Delta_v$), but once $\Delta_v$ reaches a turning point, human performance starts decreasing more rapidly. Then, we observe that spline curves obtained for certain blur and noise transformations have similar shapes, while those for dissimilar transformations start decreasing at different turning points with different slopes. More specifically, the differences between two spline curves are statistically insignificant if their $83\%$ confidence intervals overlap (Koenker et al., 1994).

Summary: By checking statistical significance with $83\%$ confidence interval for each corruption function, we empirically observed two classes of visually similar corruptions in our experiments with humans: (1) the noise class: Shot Noise, Impulse Noise, Gaussian Noise, and Uniform Noise; and (2) the blur class: Blur, Median Blur, Gaussian Blur, Glass blur, Defocus Blur. The remainder of the 14 corruptions we considered are dissimilar. See Fig. 5.

**NN Robustness for Visually Similar Corruption Functions.** Because of the central difference between humans and NNs, e.g., computational powers, it is intuitive that NNs might react completely differently to corruptions visually similar to humans, and using VCR, we can empirically analyze such difference. For example, during deployment, noise with unknown distributions (ranging from Uniform, Gaussian, Poisson etc.), can be encountered. While noise distribution does not affect humans as we showed in Fig. 5, NNs which are particularly susceptible to a certain distribution might raise safety concerns. For example, two visually similar transformations Gaussian Noise and Uniform Noise add an additional noise to the images with the Gaussian and the Uniform distribution, respectively. However, our results in Fig. 3 and Fig. 4 suggest that the distribution difference is picked up by NNs. We can observe that most models have higher *HMRI* and *MRSI* values for Uniform Noise than Gaussian Noise. For small amounts of corruption ($\Delta_v < 0.8$), the difference between the estimated $s_a$ and $s_p$ curves for both corruptions is not statistically significant, i.e., NN models perform similarly when facing small amounts of Uniform and Gaussian Noise. For $\Delta_v$ values between $[0.8..1.0]$, most visual information required for humans to recognize objects is corrupted by the noise, human performance decreases quickly, but the most robust models, e.g., DINOV2_GIANT and TIAN_DEIT-S, are able to pick up more information than humans and make reasonable recognition. When the added noise is from a uniform distribution, NN models perform better than when it is from a Gaussian distribution. Therefore, studying VCR also allows us to empirically analyze how changing the noise distribution, which would not affect humans, affects NN performance for different degrees of corruption. In the case of unknown or shifting distributions, such analysis would require human data for all distributions which is impractical and expensive. Identifying classes of visually similar corruption functions and reusing human data would significantly reduce the experiment costs.

**Identifying Visually Similar Transformations.** We provide a naive method for identifying classes of visually similar corruptions. To identify whether two corruptions are similar enough to reuse human data, the goal is to determine whether the difference between them is distinguishable to a human. This can be done through a set of relatively inexpensive experiments. Without knowing the specific

corruptions introduced to the images, participants are shown corrupted images and asked if the presented images are corrupted with the same corruption function. Presented images can be corrupted with the same or different corruption functions. Then, by repeating the experiments with different sampled images, the accuracy of distinguishing the corruptions can be calculated. We hypothesize that if the corruption functions are indistinguishable, human accuracy should be close to random. Then, since each experiment is either successfully distinguished or not, we use a binomial test to check whether the accuracy is statistically significant to not be random. Visually similar transformations included in this paper can be detected with this naive method. Our experiments showed that for each pair of transformations, results with statistical significance can be reached in less than a minute. Compared to the full set of experiments with 2,000 images and five different participants for each experiment, identifying similar transformations significantly decreased the experiment time, from approximately 5.55 hours to 5 minutes.

Limitation: Note that the results of this method can be highly dependent on the opinion of the participants; thus, it is more optimal to select participants with a normal eyesight and a basic knowledge of image corruptions. We acknowledge that this naive method cannot give the most accurate identification of visually similar transformations. For example, it is reasonable to assume that two transformations can have very different visual effects but still affect human robustness in the same way, and this case would not be detected with this method. Nevertheless, we hope that our findings will promote future investigations of how NNs and humans react differently to corruptions.

## 6 RELATED WORK

We briefly review related work on the comparison of human and NN robustness; a more extensive review of related work on robustness can be found in the appendix.

Prior studies have used human performance to study the existing differences between humans and neural networks (Firestone, 2020; Zhang et al., 2018c), to study invariant transformations (Kheradpisheh et al., 2016), to compare recognition accuracy (Ho-Phuoc, 2018; Stallkamp et al., 2012), to compare robustness against image transformations (Geirhos et al., 2019a; 2021), or to specify expected model behaviour (Hu et al., 2022). The main difference between our study and existing work, specifically, the most recent study by Geirhos et al. (2021), is three-fold: 1) we are the first to quantify robustness across the full continuous visual corruption range, thus revealing previous undetected robustness gap; 2) our experiments for obtaining human performance are designed to include more participants for measuring the *average* human robustness, resulting in more generalizable results and reduced influence of outliers; 3) we identified visually similar transformations for humans but not NNs, potentially reducing experiment costs.

## 7 CONCLUSION

In this paper, we revisit corruption robustness to consider it in relation to the wide and continuous range of corruptions to human perceptive quality, defining *visually-continuous corruption robustness* (VCR); along with two novel human-aware metrics for NN evaluation. Our results showed that **the robustness gap between human and NNs is bigger than previously detected**, especially for blur corruptions. We found that using the full and continuous range of visual change is necessary when estimating robustness, as **insufficient coverage can lead to biased results**. We also discovered classes of image corruptions that affect human perception similarly and identifying them can **help reduce the cost of measuring human robustness** and assessing disparities between human perception and computational models. In our study, we only considered the comparison of object recognition between humans and NNs; however, human and machine vision can be compared in many different ways, e.g., against neural data (Yamins et al., 2014; Kubilius et al., 2019), contrasting Gestalt effects (Kim et al., 2019), object similarity judgments (Hebart et al., 2020), or mid-level properties (Storrs et al., 2021). Nevertheless, our results give indicators for future robustness studies, and to promote further research, we provide our benchmark datasets with human performance data and our code as open source.

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
