# 8 SUPPLEMENTARY MATERIAL

## 8.1 IMPLEMENTATION AND DATA

Data and implementation can be found at https://anonymous.4open.science/r/vcr.

## 8.2 ADDITIONAL RELATED WORK

**Adversarial Robustness.** Adversarial robustness measures the worst-case performance on images with added 'small' distortions or perturbations tailored to confuse a classifier (Hendrycks & Dietterich, 2019). However, changes that can be encountered in the real-world situations are often of a much bigger range (Kar et al., 2022). Thus, in this paper, we focus on *average-case performance* over a *realistic* range of changes.

**Robustness Benchmarks.** Several robustness benchmarks have been developed. Hendrycks et al. built the IMAGENET-C and -P benchmarks for checking NN model classification robustness against common corruptions and perturbations on IMAGENET images (Hendrycks & Dietterich, 2019). They have inspired other benchmarks for different corruption functions, datasets, and tasks (Kar et al., 2022; Chattopadhyay et al., 2021; Kamann & Rother, 2021; Michaelis et al., 2019; Mintun et al., 2021; Sun et al., 2022; Yi et al., 2021). However, these benchmarks generate images by applying corruption functions with only five pre-selected values per parameter. IMAGENET-CCC (Press et al., 2023) is the only prior work targeting a more continuous range of corruptions, by using 20 pre-selected values per parameter. It does not check the coverage in terms of the effects on the images, however, which we do using a visual quality assessment (VQA) metric (VIF). Further, this work focuses on continuous changes over time for benchmarking test-time adaptation, which is very different from a general robustness benchmark, and the dataset has not been released as of writing. In contrast to all these previous works, our method randomly and uniformly samples parameter values to cover the full range of visual change that a corruption function can achieve, which is modeled and assessed for coverage using a VQA metric. Finally, our work compares robustness of NNs with humans.

**Improving Robustness.** Numerous methods for improving model robustness have been proposed, e.g., data augmentation with corrupted data (Geirhos et al., 2020; Lopes et al., 2019; Madry et al., 2018; Rusak et al., 2020), texture changes (Geirhos et al., 2019b; Hendrycks et al., 2021a), image compositions (Yun et al., 2019; Zhang et al., 2018a) and corruption functions (Yin et al., 2019; Hendrycks et al., 2020). All of these have different abilities to generalize to unseen data (Kar et al., 2022). Although improving robustness is not the focus of this paper, we show that NN robustness relative to humans can be improved through data augmentation and fine-tuning with images generated by VCR-Bench (see Sec. 8.10).

## 8.3 OVERVIEW OF VCR-BENCH

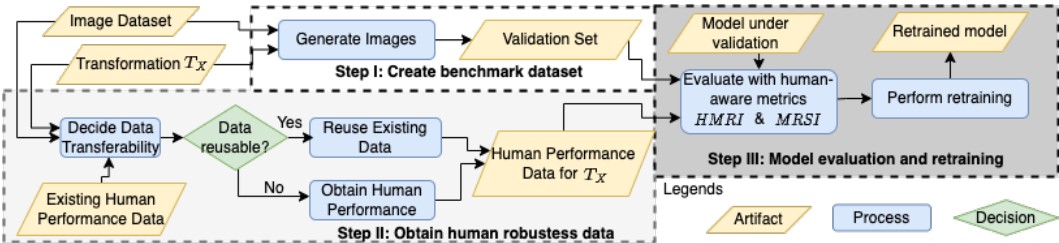

Figure 6: Our proposed method VCR-Bench for benchmarking ML robustness with humans.

Our method for benchmarking VCR (VCR-Bench) is outlined in Fig. 6. Step I generates a validation set that covers the full continuous range of visual changes. This is achieved by uniformly sampling from the entire domain of corruption function parameters. Step II obtains human robustness performance data needed to compute our two newly-proposed human-aware evaluation metrics: *Human-Relative Model Robustness Index* (HMRI) and *Model Robustness Superiority Index* (MRSI), which quantify the extent to which a NN can replicate or surpass human performance, respectively. Since measuring human performance for every single image corruption function is expensive and

impractical, we propose a method to reduce the cost by generalizing existing human performance data obtained for one corruption function to a class of corruption functions with similar visual effects. For example, images transformed with Gaussian Blur and Glass Blur have very similar visual effects on humans, unlike Motion Blur and Brightness. Thus, Gaussian Blur and Glass Blur, but not with Motion Blur and Brightness, thus they belong to the same class of similar corruption functions, and human performance data for one can be transferred to the other. Step III of VCR-Bench evaluates the model using the validation dataset and our human-aware metrics. Then it retrains the model to improve its robustness.

## 8.4 A VISUAL SUMMARY OF VCR METRICS AND THEIR ESTIMATION

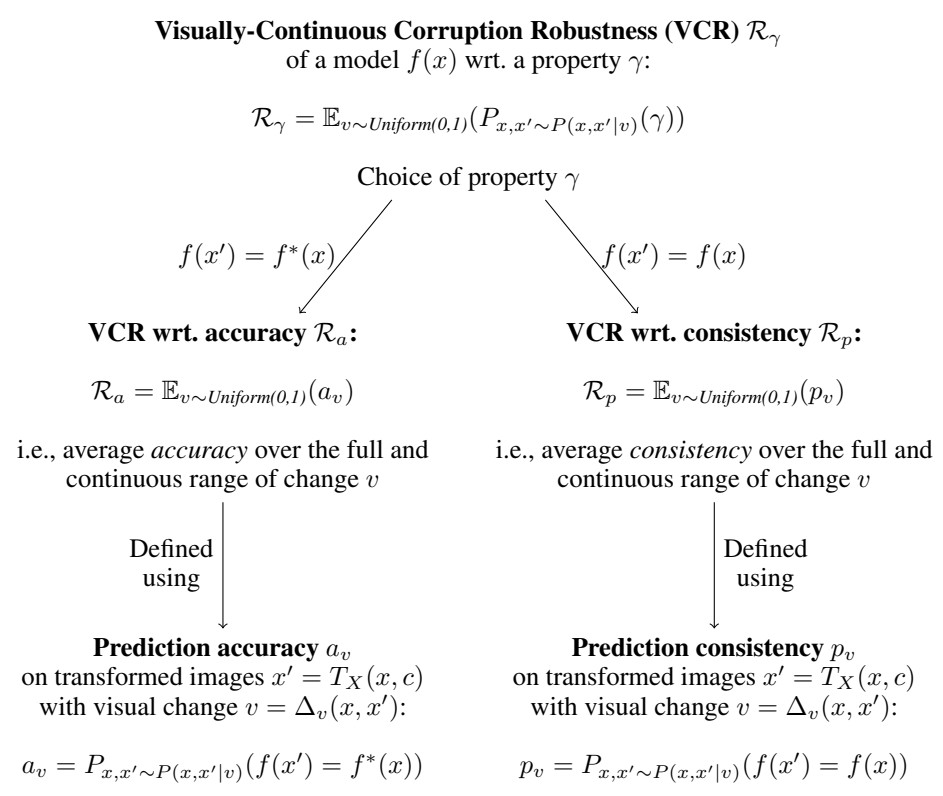

**Visually-Continuous Corruption Robustness (VCR)** $\mathcal{R}_\gamma$
of a model $f(x)$ wrt. a property $\gamma$:

$$\mathcal{R}_\gamma = \mathbb{E}_{v \sim Uniform(0,1)}(P_{x,x' \sim P(x,x'|v)}(\gamma))$$

Choice of property $\gamma$

$f(x') = f^*(x)$      $f(x') = f(x)$

**VCR wrt. accuracy** $\mathcal{R}_a$:      **VCR wrt. consistency** $\mathcal{R}_p$:

$$\mathcal{R}_a = \mathbb{E}_{v \sim Uniform(0,1)}(a_v) \qquad \mathcal{R}_p = \mathbb{E}_{v \sim Uniform(0,1)}(p_v)$$

i.e., average *accuracy* over the full and continuous range of change $v$      i.e., average *consistency* over the full and continuous range of change $v$

Defined using      Defined using

**Prediction accuracy** $a_v$
on transformed images $x' = T_X(x,c)$
with visual change $v = \Delta_v(x,x')$:
    
**Prediction consistency** $p_v$
on transformed images $x' = T_X(x,c)$
with visual change $v = \Delta_v(x,x')$:

$$a_v = P_{x,x' \sim P(x,x'|v)}(f(x') = f^*(x)) \qquad p_v = P_{x,x' \sim P(x,x'|v)}(f(x') = f(x))$$

Figure 7: Summary of VCR definitions wrt. accuracy and consistency

**Summary of VCR Definitions**. Figure 7 gives a visual summary of the VCR metrics, starting with the general definition $\mathcal{R}_\gamma$ at the top, and instantiating it for accuracy as $\mathcal{R}_a$ and consistency as $\mathcal{R}_p$. Each of them is simply the average accuracy or consistency, respectively, over the full and continuous range of visual change.

**VCR Estimation Algorithm**. Algorithm 1 gives the pseudo-code of the VCR estimation procedure described under "Testing VCR" in the main body of the paper. The algorithm takes a model $f(x)$; a transformation $T_X$ with its parameter domain $C$; an input dataset; the size $N$ of the dataset of transformed images to be generated; the visual change resolution $M$, over which the model performance will be estimated; and the minimum size $L$ of a bin to be used to estimate the performance for that bin. The input dataset consist of images $x_k \sim P_X$ for estimating VCR wrt. consistency, or images and their labels for estimating VCR wrt. accuracy. We use $M = 40$ in our experiments, which is a standard choice for calculating average precision in object detection; for example, it is used in the current version of the KITTI benchmark Geiger et al. (2013).

Our algorithm first initializes two histogram arrays to keep the counts of the tested data points and their consistent or accurate predictions, respectively, and an array to keep the performance data, with each of the three arrays having size $M$. In each iteration, the first for-loop samples an image $x$ and transformation parameter $c$, and produces a transformed image $x'$. It then computes the visual change

---

**Algorithm 1:** VCR Estimation

---

**Input:**
$\begin{cases} \text{model } f(x) \\ \text{transformation } T_X, \text{with parameter domain } C \\ \text{input dataset } \{x_k\} \text{ for consistency [or } \{(x_k, y_k)\} \text{ for accuracy]} \\ \text{generated test set size } N \\ \text{visual change resolution } M \\ \text{minimum number of points per bin } L \end{cases}$

**Output:** estimated VCR $\hat{\mathcal{R}}_p$ [or $\hat{\mathcal{R}}_a$]

Initialize histograms $count_j$ and $correct_j$ with empty counts, for all $j \in [0..M-1]$
Initialize performance data array $P_j$ with $-1$, denoting missing data points for $j$, for all
  $j \in [0..M-1]$
**for** $i \leftarrow 0$ **to** $N-1$ **do**
  draw random $x$ from $\{x_k\}$ [or $(x, y)$ from $\{(x_k, y_k)\}$]
  $c \sim Uniform(C)$
  $x' \leftarrow T_X(x, c)$
  $v \leftarrow \Delta_v(x, x')$
  $j \leftarrow \lfloor v(M-1) \rfloor$
  $count_j \leftarrow count_j + 1$
  **if** $f(x') = f(x)$ [or $f(x') = y$] **then**
    $correct_j \leftarrow correct_j + 1$
**for** $j \leftarrow 0$ **to** $M-1$ **do**
  **if** $count_j \geq L$ **then**
    $P_j \leftarrow \frac{correct_j}{count_j}$
$s \leftarrow FitMonotonicSpline(P)$
$\hat{\mathcal{R}} \leftarrow \int_0^1 s(v)dv$
**return** $\hat{\mathcal{R}}$

---

value $v$ and records the result of testing $f(x')$ in the histograms. The second for-loop computes the performance data as a relative frequency of correct predictions. A monotonic smoothing spline is fit into the performance data, and the VCR is computed as the area under the spline.

Note that this algorithm samples $c$ uniformly, which will lead to a varying number of performance samples per point in the performance data array $P$. As already discussed, the number of performance samples impacts the performance estimate uncertainty at this point, and in an extreme case some of the $\Delta_v$ bins in $P_i$ may be even empty (i.e., have value -1). These missing points are mitigated by fitting the spline over the entire $\Delta_v$ range, while anchoring it with known values for the first and last bins. In particular, the accuracy spline $s_a$ always starts at the left with the accuracy for clean images, and the consistency spline $s_p$ starts with 1 for models (assuming deterministic NNs).

A possible approach to obtain a sample set with a more uniform coverage of $\Delta_v$ would be to (1) fit a strictly monotonic spline into $(c, \Delta_v)$ values obtained from $c \sim Uniform(C)$ as in Alg. 1, (2) take a set of samples $\Delta_v \sim Uniform(0, 1)$, (3) map the latter to a new sample from $C$ using the inverted spline, and repeat these steps now using the new sample from $C$. These steps would need to be run iteratively until a sufficient coverage is obtained. Such an algorithm would be computationally expensive, however.

**Auxiliary VCR metrics to compute HMRI and MSRI**. In order to compare model and human performance, VCR wrt. consistency or accuracy is estimated using Alg. 1 for model and human performance data, as illustrated by the yellow ($A^h$) and blue ($A^m$) areas Fig. 8, respectively, where both blue and orange area also include the green area representing their overlap. Additionally, the VCR lead of humans over a model $A^{h>m}$, the girded area in Fig. 8, and the VCR lead of a model over humans $A^{m>h}$, the striped area in Fig. 8, are estimated. The definitions of the four auxiliary metrics are summarized in Tab. 2

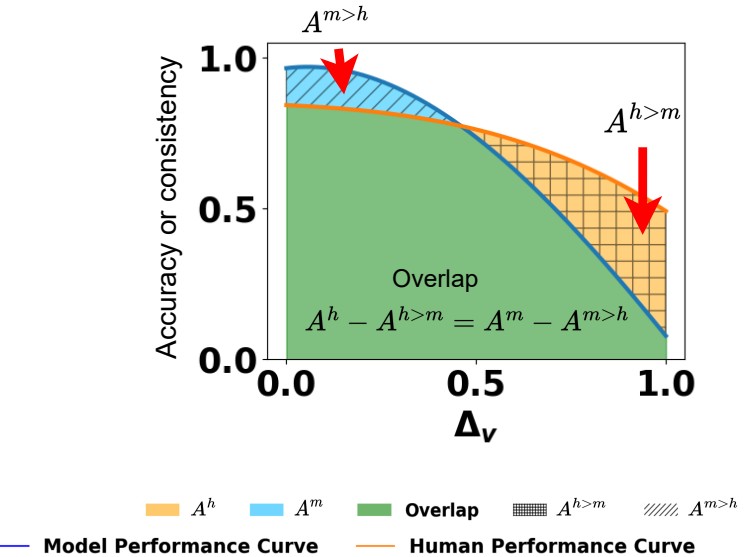

Figure 8: Visualization of auxiliary metrics for model vs. human performance

| Auxiliary metric (cf. Fig. 8) | Definition |
|---|---|
| **VCR of humans** wrt. a property $\gamma$, estimated as an area under performance curve $A_\gamma^h$ | $\hat{\mathcal{R}}_\gamma^h = A_\gamma^h = \int_0^1 s_\gamma^h(v)dv$ |
| **VCR of a model** $f(x)$ wrt. a property $\gamma$, estimated as an area under performance curve $A_\gamma^m$ | $\hat{\mathcal{R}}_\gamma^m = A_\gamma^m = \int_0^1 s_\gamma^m(v)dv$ |
| **VCR lead of humans over a model** $f(x)$ wrt. a property $\gamma$, estimated as a difference area $A_\gamma^{h>m}$ | $\hat{\mathcal{R}}_\gamma^{h>m} = A_\gamma^{h>m}$ $= \int_0^1 \max(0, s_\gamma^h(v) - s_\gamma^m(v))dv$ |
| **VCR lead of a model** $f(x)$ **over humans** wrt. a property $\gamma$, estimated as a difference area $A_\gamma^{m>h}$ | $\hat{\mathcal{R}}_\gamma^{m>h} = A_\gamma^{m>h}$ $= \int_0^1 \max(0, s_\gamma^m(v) - s_\gamma^h(v))dv$ |

Table 2: Summary of auxiliary metrics for defining HMRI and MRSI

**HMRI and MSRI Definitions**. Finally, the auxiliary metrics are used to define: (i) Human-Relative Model Robustness Index (HMRI), which characterizes the human lead in VCR over the model; and (ii) Model Robustness Superiority Index (MRSI), which characterizes the model lead in VCR over humans. Their definitions are summarized in Tab. 3.

| Human-model comparison metric | Definition |
|---|---|
| **Human-Relative Model Robustness Index (HMRI)** of a model $f(x)$ wrt. a property $\gamma$ | $HMRI_\gamma = \frac{A_\gamma^h - A_\gamma^{h>m}}{A_\gamma^h} = 1 - \frac{A_\gamma^{h>m}}{A_\gamma^h}$ |
| **Model Robustness Superiority Index (MRSI)** of a model $f(x)$ wrt. a property $\gamma$ | $MRSI_\gamma = \frac{A_\gamma^{m>h}}{A_\gamma^m}$ |

Table 3: HMRI and MRSI definitions (using auxiliary metrics from Tab. 2)

## 8.5 VISUAL CHANGE IN MORE DETAILS

**Background: Image Quality Assessment (IQA)**. IQA metrics serve as quantitative measures of human objective image quality (Wang et al., 2004). By comparing the original image and the transformed image, IQA metrics automatically estimate the perceived image quality by evaluating the perceptual "distance" between the two images (Sheikh & Bovik, 2006). This "distance" differs from simple pixel distance and varies depending on the specific IQA metric used.

One such metric is VIF (Visual Information Fidelity) (Sheikh & Bovik, 2006), which evaluates the fidelity of information by analyzing the statistical properties of natural scenes within the images. VIF returns a value between 0 and 1 if the changes degrade perceived image quality, with 1 indicating the perfect quality compared to the original image; and it returns a value $> 1$ if the changes enhances image quality (Sheikh & Bovik, 2006). More precisely, VIF defines the visual quality of a distorted image as a ratio of the amount of information a human can extract from the distorted image versus the original reference image. The method models statistically (i) images in the wavelet domain with coefficients drawn from a Gaussian scale mixture, (ii) distortions as attenuation and additive Gaussian noise in the wavelet domain, and (iii) the human visual system (HVS) as additive white Gaussian noise in each sub-band of the wavelet decomposition. The amount of information that a human can extract from the distorted image is measured as the mutual information between the distorted image and the output of the HVS model for that image. Similarly, the amount of information that a human can extract from the reference image is measured as the mutual information between the reference image and the output of the HVS model for that image. Empirical studies have shown that VIF aligns closely with human opinions when compared to other IQA metrics (Sheikh et al., 2006).

We choose VIF, since it is well-established, computationally efficient, applicable to our transformations, and still performing competitively compared to newer metrics. More recent research has explored the use of feature spaces computed by deep NNs as a basis to define IQA metrics (e.g., LPIPS (Zhang et al., 2018b) and DISTS (Ding et al., 2022)). Even though these metrics may be applicable to a wider class of transformations than VIF, including those that affect both structure and textures, their scope may depend on the training datasets in potentially unpredictable ways. On the other hand, the scope of VIF is well-defined based on the metric's mathematical definition. In particular, VIF is suitable for evaluating corruption functions that can be locally described as a combination of signal attenuation and additive Gaussian noise in the sub-bands of the wavelet domain (Sheikh & Bovik, 2006). The transformations in our experiments are local corruptions that are well within this scope. Moreover, VIF performs still competitively when compared to even the newer DNN-based metrics across multiple datasets (e.g., see Table 1 in (Ding et al., 2022)). However, future work should explore VCR using other IQA metrics.

**Visual Change ($\Delta_v$)**. The metric ($\Delta_v$) defined using the IQA metric VIF, as shown in Def. 3, is proposed by Hu et al. (Hu et al., 2022) to quantitatively measure the amount of visual changes in the images perceived by human observers.

**Definition 3.** Let an image $x$, an applicable corruption function $T_X$ with a parameter domain $C$ and a parameter $c \in C$, s.t. $x' = T_X(x, c)$ be given. Visual change $\Delta_v(x, x')$ is a function defined as follows:

$$\begin{cases} 0 & \text{If VIF}(x, x') > 1 \\ 1 - \text{VIF}(x, x') & \text{Otherwise} \end{cases}$$

$\Delta_v$ returns a value between 0 and 1, with 0 indicating no degradation to visual quality and 1 indicating all visual information in the original image has been changed. The first case of $\Delta_v$ corresponds to changes that enhance the visual quality (when $\text{VIF}(x, x') > 1$), indicating changes do not impact human recognition of the images negatively, hence $\Delta_v = 0$. The other case deals with visible changes that degrade visual quality. Since VIF returns 1 for perfect quality compared to the original image, the degradation is one minus the image quality score.

Example: In Fig. 9, the visual change of the original image Fig. 9a is 0, since no changes are applied; and Fig. 9b has minimal frost added, which caused minimal change in visual quality so $\Delta_v = 0.005$; and Fig. 9c and Fig. 9d have more frost and thus higher $\Delta_v$ values 0.71 and 0.96, respectively.

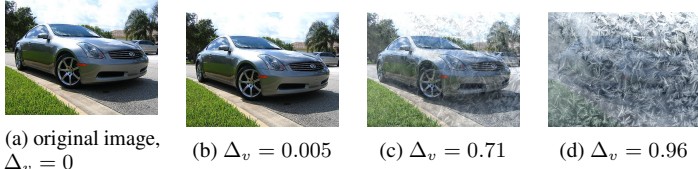

(a) original image, $\Delta_v = 0$  (b) $\Delta_v = 0.005$  (c) $\Delta_v = 0.71$  (d) $\Delta_v = 0.96$

Figure 9: Examples of images from Imagenet (Russakovsky et al., 2015) with different levels of added frost.

## 8.6 CORRUPTION FUNCTIONS INCLUDED IN OUR STUDY

The 14 image corruption functions discussed in the paper are shown in Fig. 10. All corruption functions are demonstrated with the same original image.

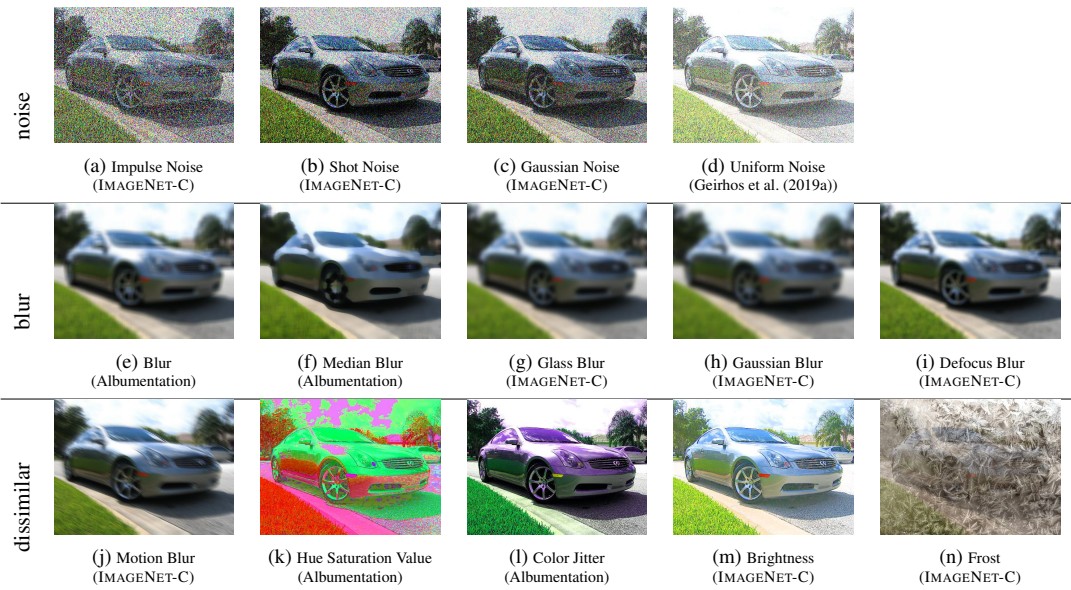

(a) Impulse Noise (IMAGENET-C)  (b) Shot Noise (IMAGENET-C)  (c) Gaussian Noise (IMAGENET-C)  (d) Uniform Noise (Geirhos et al. (2019a))

(e) Blur (Albumentation)  (f) Median Blur (Albumentation)  (g) Glass Blur (IMAGENET-C)  (h) Gaussian Blur (IMAGENET-C)  (i) Defocus Blur (IMAGENET-C)

(j) Motion Blur (IMAGENET-C)  (k) Hue Saturation Value (Albumentation)  (l) Color Jitter (Albumentation)  (m) Brightness (IMAGENET-C)  (n) Frost (IMAGENET-C)

Figure 10: Image corruption functions discussed in the paper grouped by classes of similar corruption functions.

## 8.7 COMPARISON OF $\Delta_v$ DISTRIBUTION

In Fig. 11 below we compare the $\Delta_v$ distribution of validation images from IMAGENET-C and those generated by our benchmark. We include all 9 corruption functions shared between IMAGENET-C and our benchmark. Note that all of our images are generated by sampling uniformly in the parameter domain, while IMAGENET-C images are generated with 5 pre-selected parameter values. We can observe two major differences in the distributions. First we can see that because of difference in the parameter values used, the $\Delta_v$ distributions between IMAGENET-C and our benchmark peak at different values. For example, for Brightness in Fig. 11a and Fig. 11b, most IMAGENET-C images have $\Delta_v$ values between $0.4$ to $0.8$, while most VCR-Bench images are between $0.6$ and $0.9$; a similar observation holds for Defocus Blur and Gaussian Blur. Second, we notice that IMAGENET-C images cannot cover all $\Delta_v$ values. Specifically, Fig. 11c for Defocus Blur shows that IMAGENET-C validation set does not contain images with $\Delta_v$ greater than $0.8$ and less than $0.2$. The same can be observed for all corruption functions shown in Fig. 11. These two differences indicate that, when considering the full range of visual changes that a corruption function can incur, using IMAGENET-C can lead to biased results.

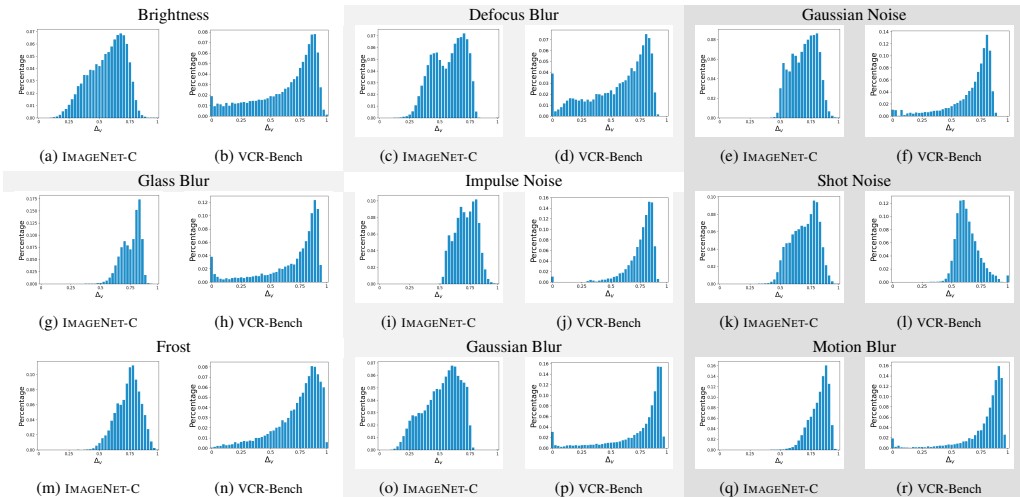

Figure 11: Comparison of $\Delta_v$ distribution between IMAGENET-C and VCR-Bench. The figures are histograms, where x-axis is $\Delta_v$, y-axis is image count.

## 8.8 MODELS INCLUDED IN OUR STUDY

Table 4 summarizes the models included in our study. We have selected a wide range of architectures (different CNN and transformer architectures) and training methods (supervised, adversarial, semi-weakly, and self-supervised), including dinov2_giant (Oquab et al., 2023), which is on the top of the IMAGENET-C leader board as of writing.

| Model | Architecture | Training Method | Source |
|---|---|---|---|
| NoisyMix | ResNet-50 | Supervised | (Erichson et al., 2022) |
| NoisyMix_new | ResNet-50 | Supervised | (Erichson et al., 2022) |
| SIN | ResNet-50 | Supervised | (Geirhos et al., 2019b) |
| SIN_IN | ResNet-50 | Supervised | (Geirhos et al., 2019b) |
| SIN_IN_IN | ResNet-50 | Supervised | (Geirhos et al., 2019b) |
| HMany | ResNet-50 | Supervised | (Hendrycks et al., 2021a) |
| HAugMix | ResNet-50 | Supervised | (Hendrycks et al., 2020) |
| Standard_R50 | ResNet-50 | Supervised | (He et al., 2016) |
| AlexNet | AlexNet | Supervised | (Krizhevsky et al., 2012) |
| Tian_DeiT-S | DeiT Small | Supervised | (Tian et al., 2022) |
| Tian_DeiT-B | DeiT Base | Supervised | (Tian et al., 2022) |
| Do_50_2_Linf | WideResNet-50-2 | Adversarial | (Salman et al., 2020) |
| Liu_Swin-L | Swin-L | Adversarial | (Liu et al., 2023) |
| Liu_ConvNeXt-L | ConvNeXt-L | Adversarial | (Singh et al., 2023) |
| Singh_ConvNeXt-L-ConvStem | ConvNeXt-L + ConvStem | Adversarial | (Singh et al., 2023) |
| swsl_resnet18 | ResNet-18 | Semi-weakly sup. | (Yalniz et al., 2019) |
| swsl_resnext101_32x16d | ResNext-101 | Semi-weakly sup. | (Yalniz et al., 2019) |
| clip | Clip | Supervised | (Radford et al., 2021) |
| dinov2_giant | ViT | Self-supervised | (Oquab et al., 2023) |

Table 4: Summary of the models included in our study

## 8.9 EXTRA EVALUATION RESULTS

### 8.9.1 PREDICTION SIMILARITY OF VISUALLY SIMILAR CORRUPTION FUNCTIONS

In the paper, to check that human robustness data is transferable between two similar corruption functions, we checked whether the 83% confidence interval of the spine curves $s_a^h$ and $s_p^h$ for similar corruption functions overlap. The results for $s_a^h$ in Fig. 5. We also include results for $s_p^h$ in Fig. 12. We can observe that, similar to $s_a^h$, $s_p^h$ for similar corruption functions are similar, thus human data is transferable.

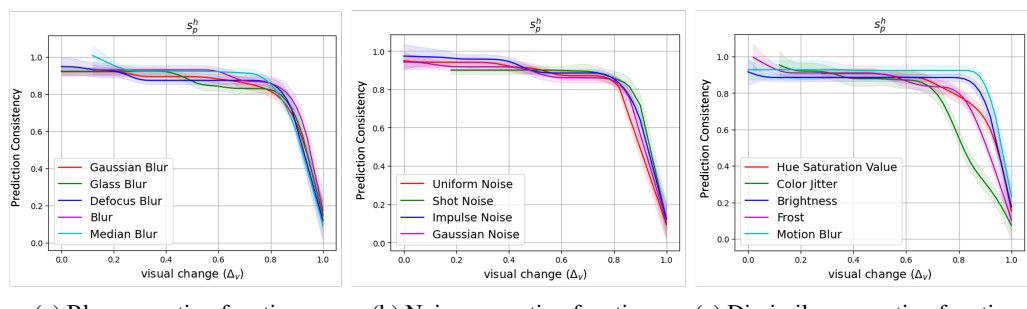

(a) Blur corruption functions  (b) Noise corruption functions  (c) Dissimilar corruption functions

Figure 12: Comparing human performance spline curves $s_p^h$ for similar and dissimilar corruption functions. For each curve, the coloured region around the curve is the $83\%$ confidence interval used for comparison of similarity (Koenker et al., 1994).

### 8.9.2 CO2 EMISSION

CO2 Emission is calculated as `CO2 emissions (kg) = (Power consumption in kilowatts) x (Daily usage time in hours) x (Emissions factor in kgCO2/kWh)`

Our carbon intensity is around 25 g/kWh. During benchmark dataset generation, there is no GPU usage, and the CPU usage is 200 W. Each corruption function takes around 1.5 hour to generate a dataset with 50,000 images. During evaluation, the CPU power usage is around 160 W; and GPU power usage ranges between 50-170 W depending on the model. Each evaluation takes 30-60 minutes, depending on the corruption function type. Let's assume the power usage of other components is 50 W in total. If we assume the total power usage is $((200 + 50) \times 1.5 + (170 + 160 + 50))/1,000 = 0.755$ kWh for each experiment, the CO2 emission is $0.755 \times 25 = 18.875$ g for each experiment (corruption function type).

### 8.10 TRAINING WITH DATA AUGMENTATION

We show a small retraining example for demonstrating the usefulness of our benchmark in improving VCR. The retraining process was carried out by fine-tuning all parameters of the image classification model. The training dataset was generated from a subset sampled from the IMAGENET (Russakovsky et al., 2015) training set with a size of around 12,000. For optimization, we leveraged the most basic stochastic gradient descent with learning rate=0.001 and momentum=0.9. We utilized Cross-Entropy Loss as the loss function, given its effectiveness in classification tasks. The number of epochs depends on the model. 5 epochs is usually enough to show some progress. The training details can also be found in the codebase: https://anonymous.4open.science/r/vcr.

The state-of-the-art NNs are already optimized for the corruption functions included in IMAGENET-C; however, as shown in Tbl. 1, for certain corruption functions, such as Motion Blur, Frost and Glass Blur, IMAGENET-C images do not cover a wide range of visual changes, leaving room for robustness improvement. In Tbl. 5 and Tbl. 6, we demonstrate results for NNs SIN (Geirhos et al., 2019b) and Standard_R50 (Croce et al., 2021) for these corruption functions, the rest can be found in the codebase.

| corruption function | Before Retraining | | | | | | After Retraining | | | | | |
| | Accuracy | | | Prediction similarity | | | Accuracy | | | Prediction similarity | | |
| | $\mathcal{R}_a$ | HMRI | MRSI | $\mathcal{R}_p$ | HMRI | MRSI | $\hat{\mathcal{R}}_a$ | HMRI | MRSI | $\hat{\mathcal{R}}_p$ | HMRI | MRSI |
|---|---|---|---|---|---|---|---|---|---|---|---|---|
| Median Blur | 0.532 | 0.635 | 0.000 | 0.573 | 0.673 | 0.000 | **0.694** | **0.828** | **0.003** | 0.728 | 0.854 | 0.001 |
| Frost | 0.429 | 0.521 | 0.011 | 0.473 | 0.572 | 0.012 | **0.575** | **0.690** | **0.025** | 0.678 | 0.804 | 0.031 |
| Glass Blur | 0.468 | 0.569 | 0.003 | 0.502 | 0.603 | 0.003 | **0.647** | **0.770** | **0.024** | 0.744 | 0.866 | 0.034 |

Note: all numbers are rounded.

Table 5: VCR comparison before and after retraining for Standard_R50 (Croce et al., 2021). Red indicates improvement.

| corruption function | Before Retraining | | | | | | After Retraining | | | | | |
| | Accuracy | | | Prediction similarity | | | Accuracy | | | Prediction similarity | | |
| | $\mathcal{R}_a$ | HMRI | MRSI | $\mathcal{R}_p$ | HMRI | MRSI | $\hat{\mathcal{R}}_a$ | HMRI | MRSI | $\hat{\mathcal{R}}_p$ | HMRI | MRSI |
|---|---|---|---|---|---|---|---|---|---|---|---|---|
| Median Blur | 0.522 | 0.624 | 0.00 | 0.605 | 0.710 | 0.00 | **0.650** | **0.774** | **0.004** | **0.729** | **0.852** | **0.004** |
| Frost | 0.423 | 0.512 | 0.015 | 0.513 | 0.618 | 0.016 | **0.517** | **0.625** | **0.016** | **0.647** | **0.768** | **0.031** |
| Glass Blur | 0.334 | 0.407 | 0.000 | 0.397 | 0.478 | 0.000 | **0.572** | **0.687** | **0.016** | **0.684** | **0.809** | **0.018** |

Note: all numbers are rounded.

Table 6: VCR comparison before and after retraining for SIN (Geirhos et al., 2019b). Red indicates improvement.

### 8.10.1 VCR EVALUATION

In the main body of the paper, we have compared VCR robustness results with IMAGENET-C on Gaussian Noise, and we presented the assessing VCR in relation to human performance with our human-aware metrics *HMRI* and *MRSI* for Gaussian Noise and Shot Noise. Below, we first include the comparison between VCR and IMAGENET-C for all IMAGENET-C 9 corruption functions we studied. Then, include detailed evaluation results with our human-aware metrics for all 12 other corruption functions we studied.

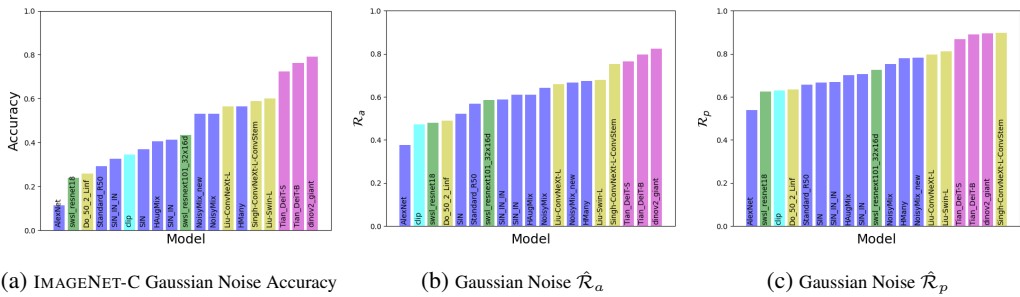

(a) IMAGENET-C Gaussian Noise Accuracy      (b) Gaussian Noise $\hat{\mathcal{R}}_a$      (c) Gaussian Noise $\hat{\mathcal{R}}_p$

Figure 13: Comparison between IMAGENET-C and VCR with Gaussian Noise.

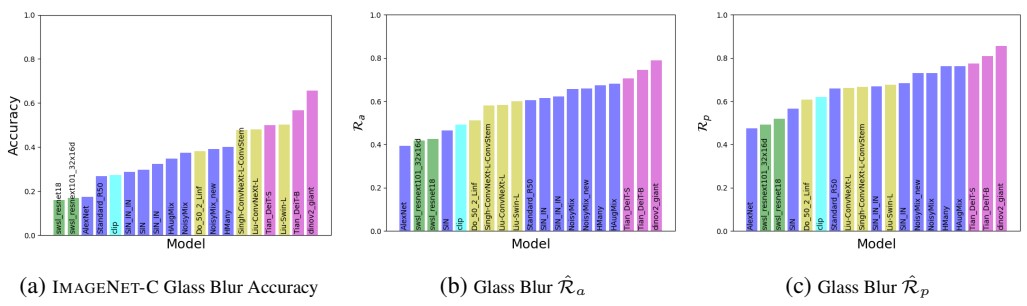

(a) IMAGENET-C Glass Blur Accuracy      (b) Glass Blur $\hat{\mathcal{R}}_a$      (c) Glass Blur $\hat{\mathcal{R}}_p$

Figure 14: Comparison between IMAGENET-C and VCR with Glass Blur.

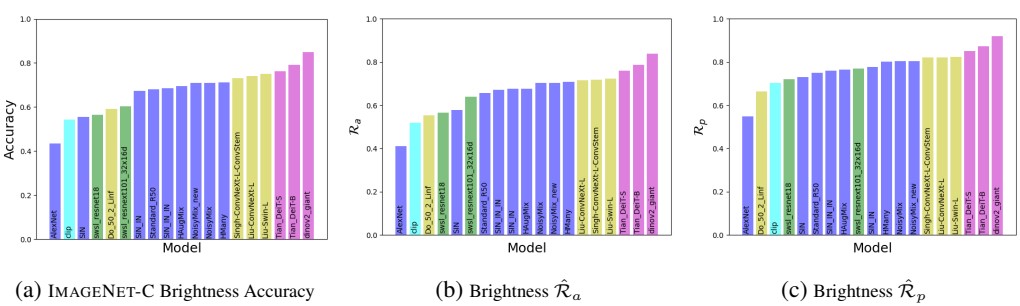

(a) IMAGENET-C Brightness Accuracy      (b) Brightness $\hat{\mathcal{R}}_a$      (c) Brightness $\hat{\mathcal{R}}_p$

Figure 15: Comparison between IMAGENET-C and VCR with Brightness.

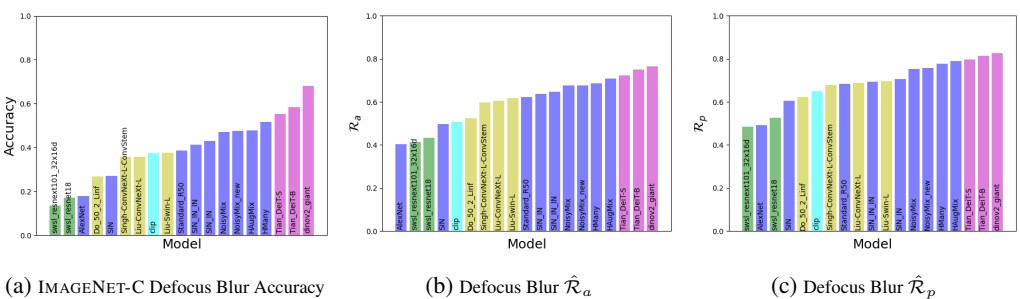

(a) IMAGENET-C Defocus Blur Accuracy      (b) Defocus Blur $\hat{\mathcal{R}}_a$      (c) Defocus Blur $\hat{\mathcal{R}}_p$

Figure 16: Comparison between IMAGENET-C and VCR with Defocus Blur.

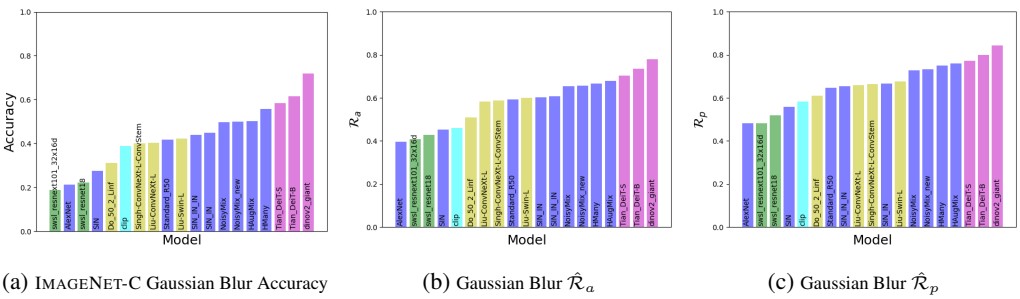

(a) IMAGENET-C Gaussian Blur Accuracy     (b) Gaussian Blur $\hat{\mathcal{R}}_a$     (c) Gaussian Blur $\hat{\mathcal{R}}_p$

Figure 17: Comparison between IMAGENET-C and VCR with Gaussian Blur.

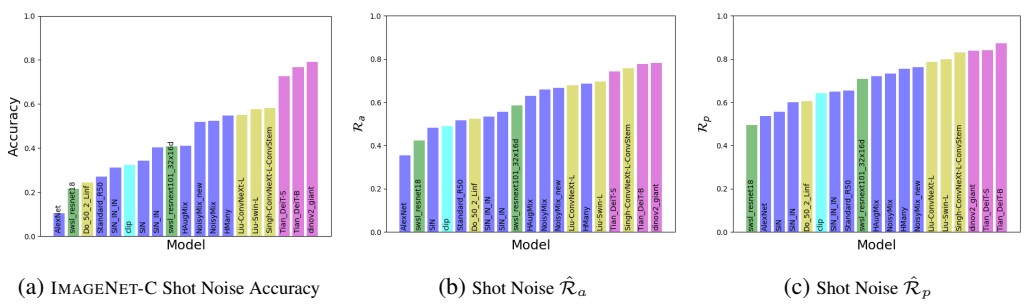

(a) IMAGENET-C Shot Noise Accuracy     (b) Shot Noise $\hat{\mathcal{R}}_a$     (c) Shot Noise $\hat{\mathcal{R}}_p$

Figure 18: Comparison between IMAGENET-C and VCR with Shot Noise.

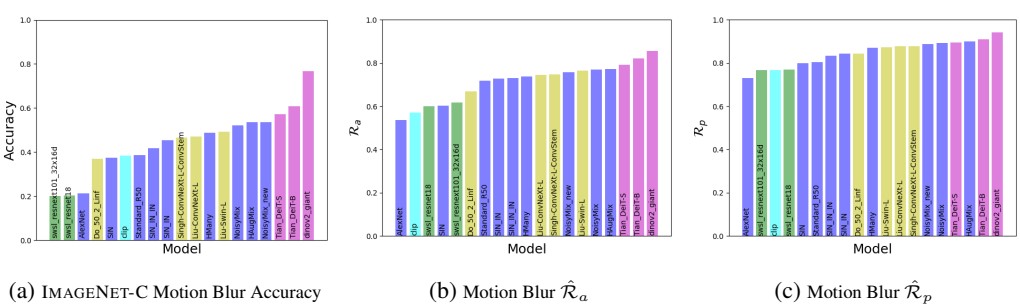

(a) IMAGENET-C Motion Blur Accuracy     (b) Motion Blur $\hat{\mathcal{R}}_a$     (c) Motion Blur $\hat{\mathcal{R}}_p$

Figure 19: Comparison between IMAGENET-C and VCR with Motion Blur.

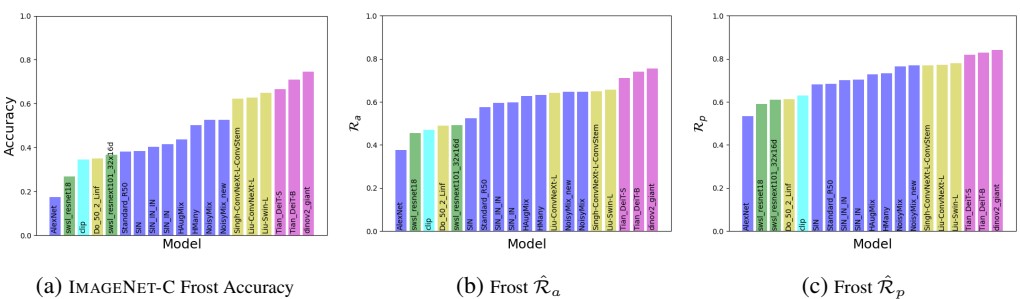

(a) IMAGENET-C Frost Accuracy     (b) Frost $\hat{\mathcal{R}}_a$     (c) Frost $\hat{\mathcal{R}}_p$

Figure 20: Comparison between IMAGENET-C and VCR with Frost.

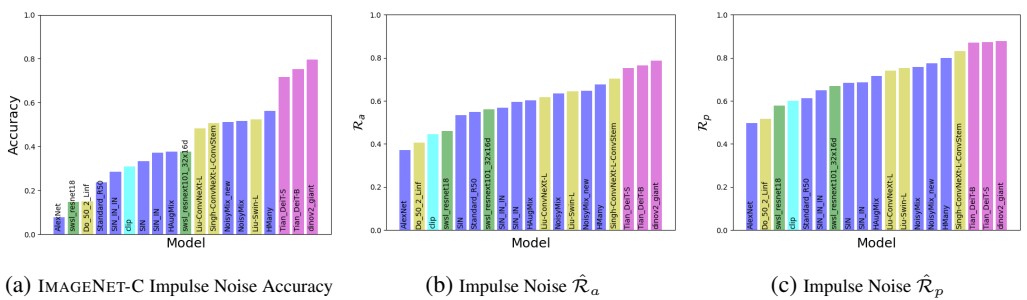

(a) IMAGENET-C Impulse Noise Accuracy

(b) Impulse Noise $\hat{\mathcal{R}}_a$

(c) Impulse Noise $\hat{\mathcal{R}}_p$

Figure 21: Comparison between IMAGENET-C and VCR with Impulse Noise.

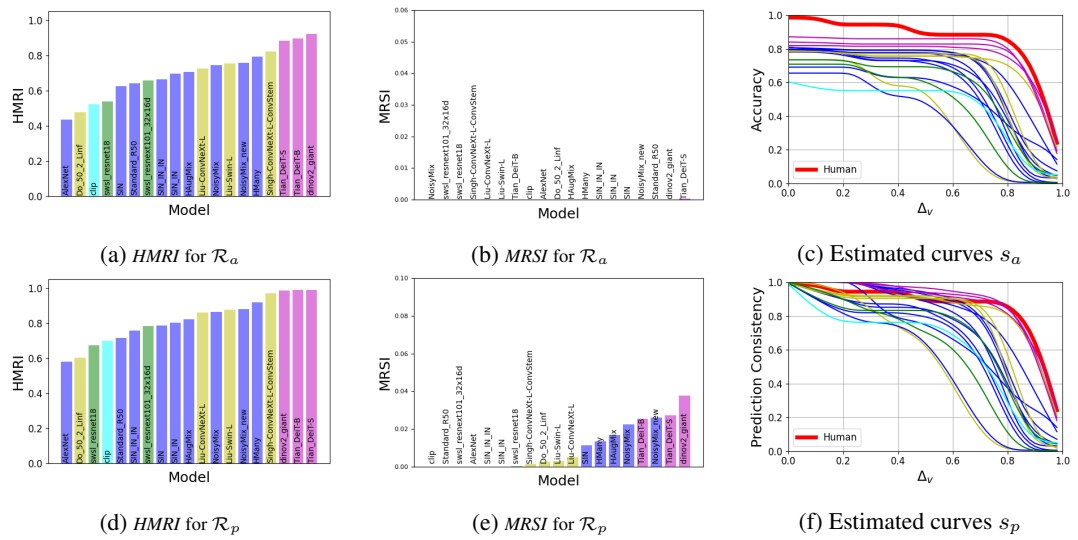

(a) HMRI for $\mathcal{R}_a$

(b) MRSI for $\mathcal{R}_a$

(c) Estimated curves $s_a$

(d) HMRI for $\mathcal{R}_p$

(e) MRSI for $\mathcal{R}_p$

(f) Estimated curves $s_p$

Figure 22: VCR evaluation results for Impulse Noise.

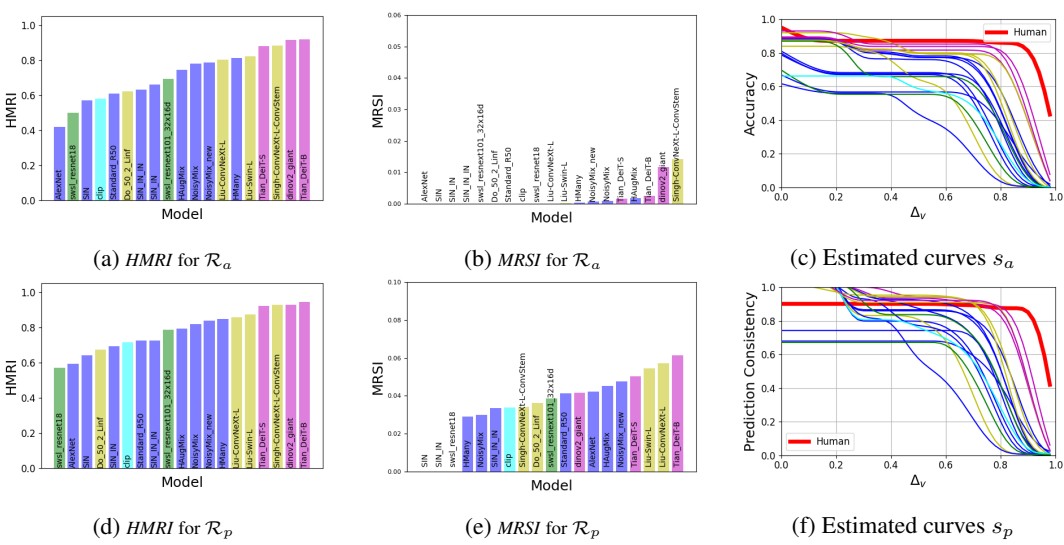

(a) HMRI for $\mathcal{R}_a$

(b) MRSI for $\mathcal{R}_a$

(c) Estimated curves $s_a$

(d) HMRI for $\mathcal{R}_p$

(e) MRSI for $\mathcal{R}_p$

(f) Estimated curves $s_p$

Figure 23: VCR evaluation results for Shot Noise.

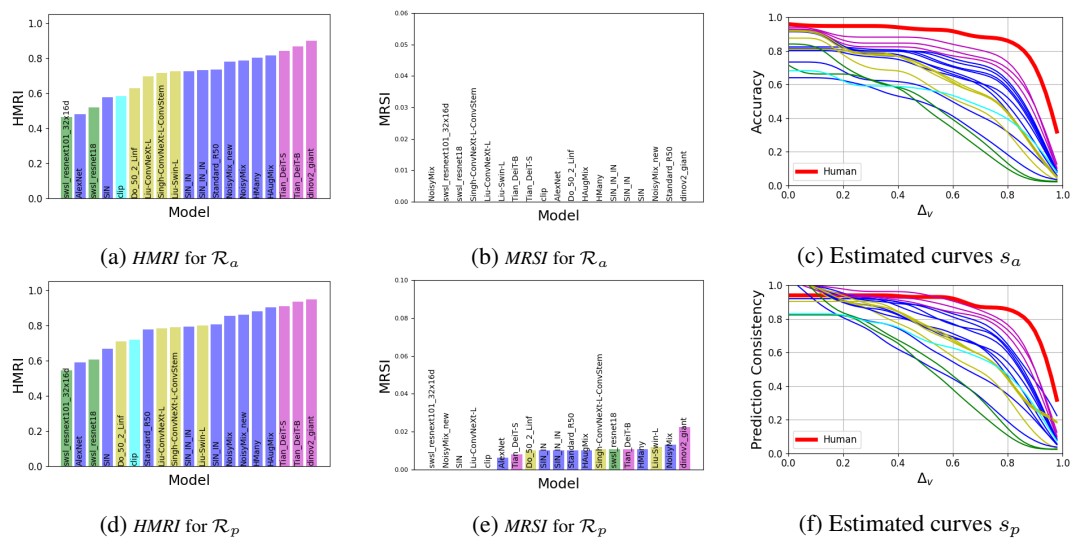

Figure 24: VCR evaluation results for Blur.

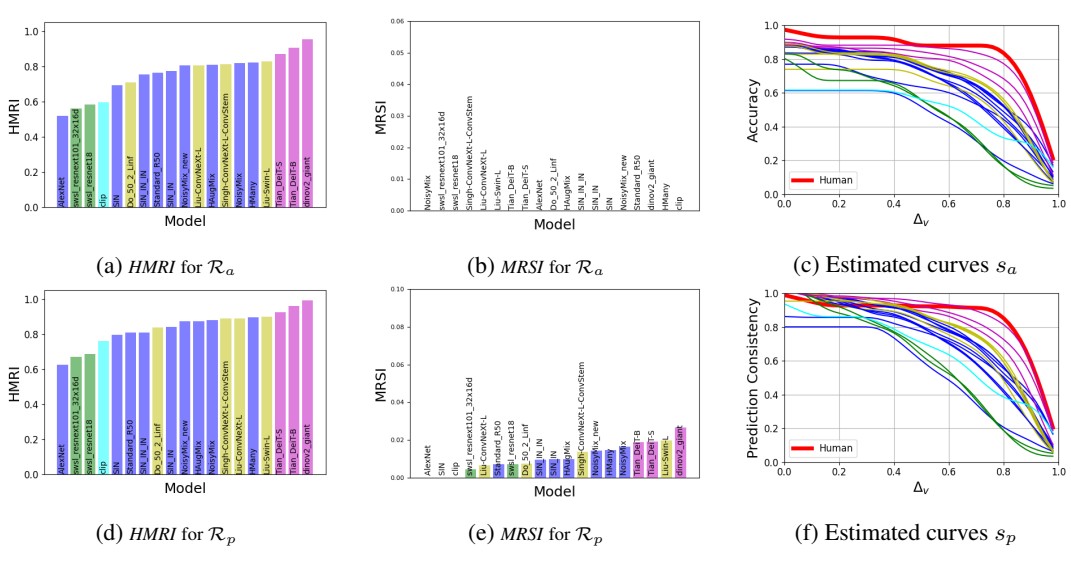

Figure 25: VCR evaluation results for Median Blur.

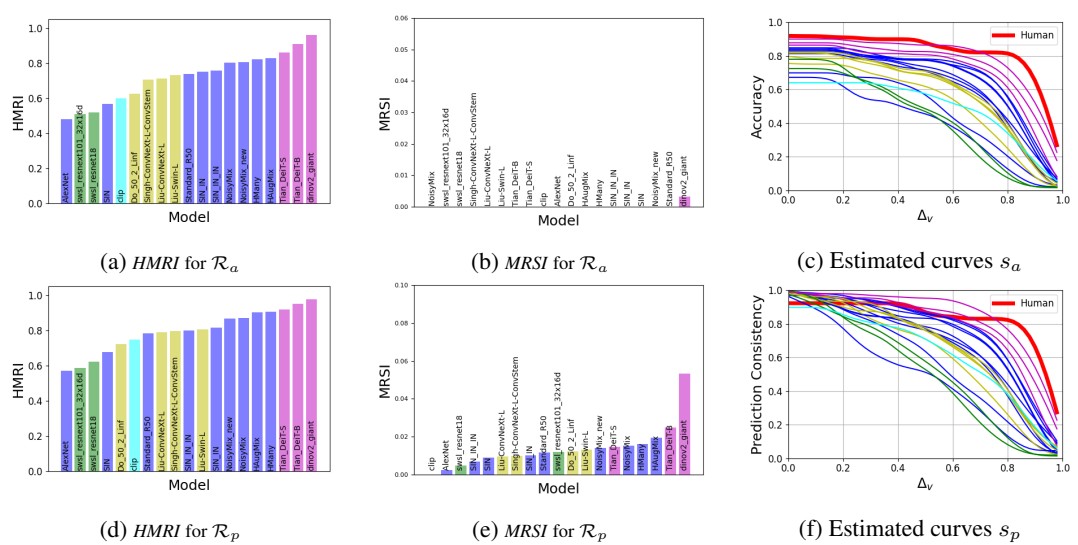

(a) *HMRI* for $\mathcal{R}_a$     (b) *MRSI* for $\mathcal{R}_a$     (c) Estimated curves $s_a$

(d) *HMRI* for $\mathcal{R}_p$     (e) *MRSI* for $\mathcal{R}_p$     (f) Estimated curves $s_p$

Figure 26: VCR evaluation results for Glass Blur.

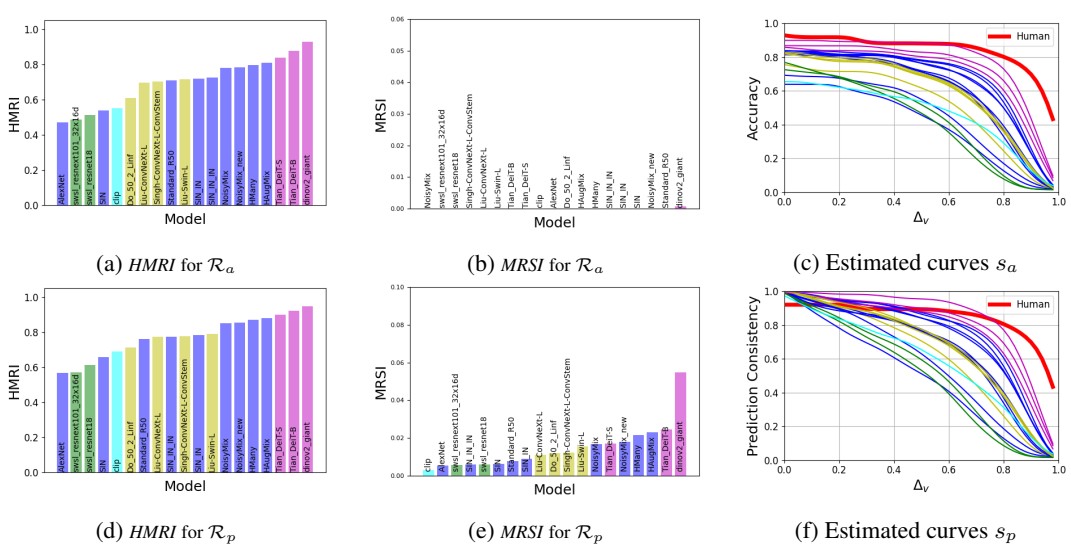

(a) *HMRI* for $\mathcal{R}_a$     (b) *MRSI* for $\mathcal{R}_a$     (c) Estimated curves $s_a$

(d) *HMRI* for $\mathcal{R}_p$     (e) *MRSI* for $\mathcal{R}_p$     (f) Estimated curves $s_p$

Figure 27: VCR evaluation results for Gaussian Blur.

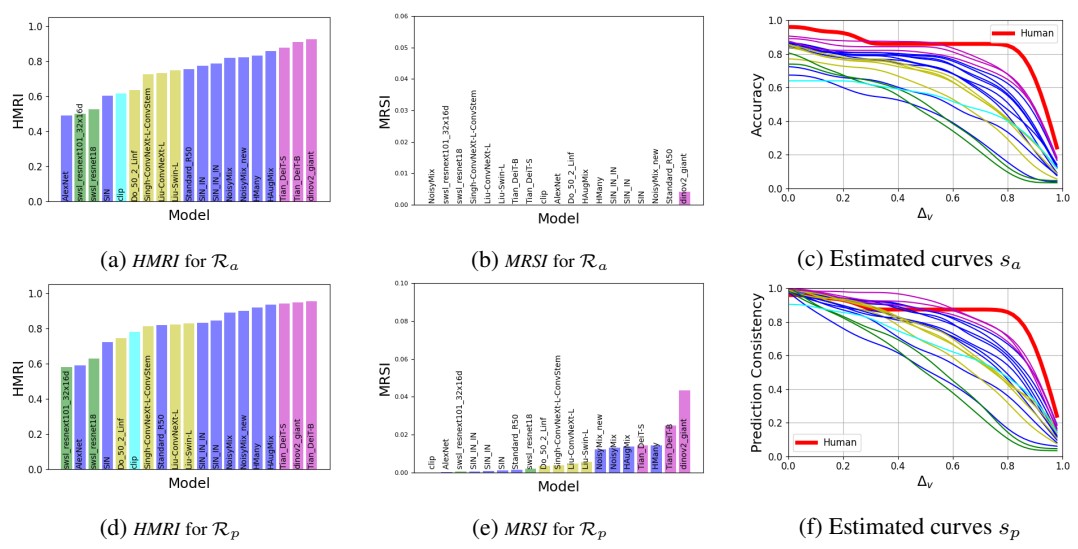

(a) *HMRI* for $\mathcal{R}_a$

(b) *MRSI* for $\mathcal{R}_a$

(c) Estimated curves $s_a$

(d) *HMRI* for $\mathcal{R}_p$

(e) *MRSI* for $\mathcal{R}_p$

(f) Estimated curves $s_p$

Figure 28: VCR evaluation results for Defocus Blur.

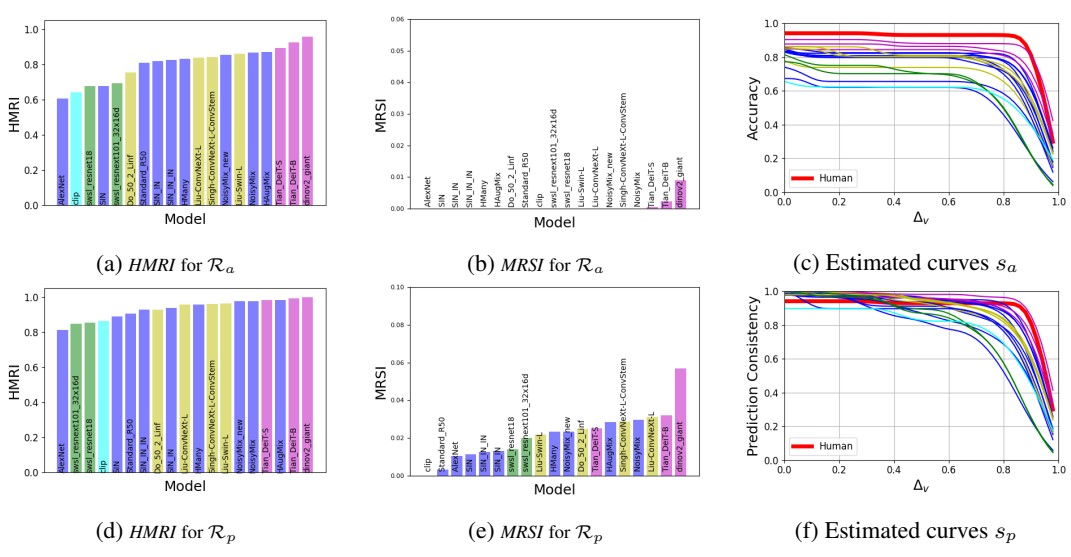

(a) *HMRI* for $\mathcal{R}_a$

(b) *MRSI* for $\mathcal{R}_a$

(c) Estimated curves $s_a$

(d) *HMRI* for $\mathcal{R}_p$

(e) *MRSI* for $\mathcal{R}_p$

(f) Estimated curves $s_p$

Figure 29: VCR evaluation results for Motion Blur.

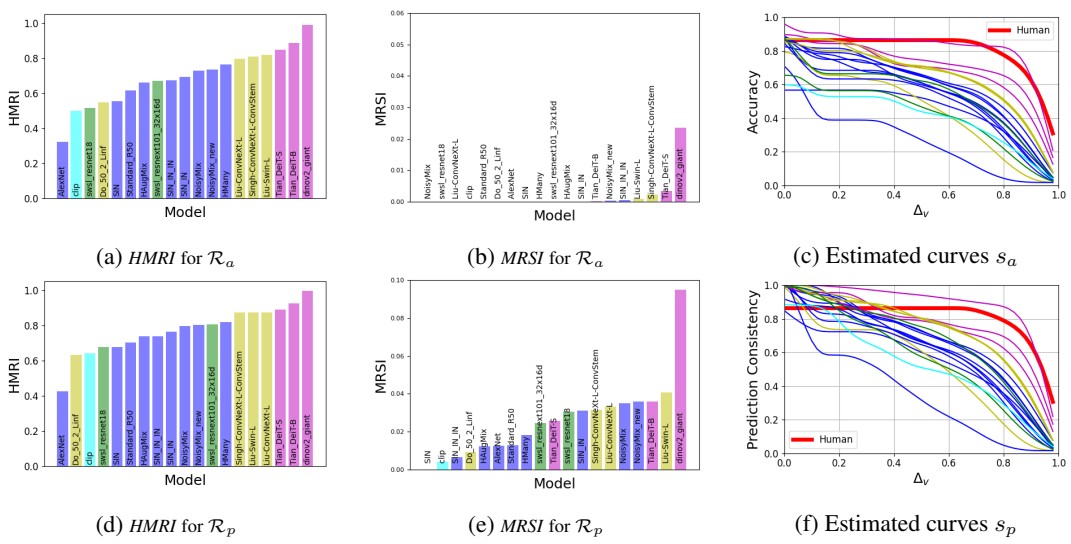

Figure 30: VCR evaluation results for Hue Saturation Value.

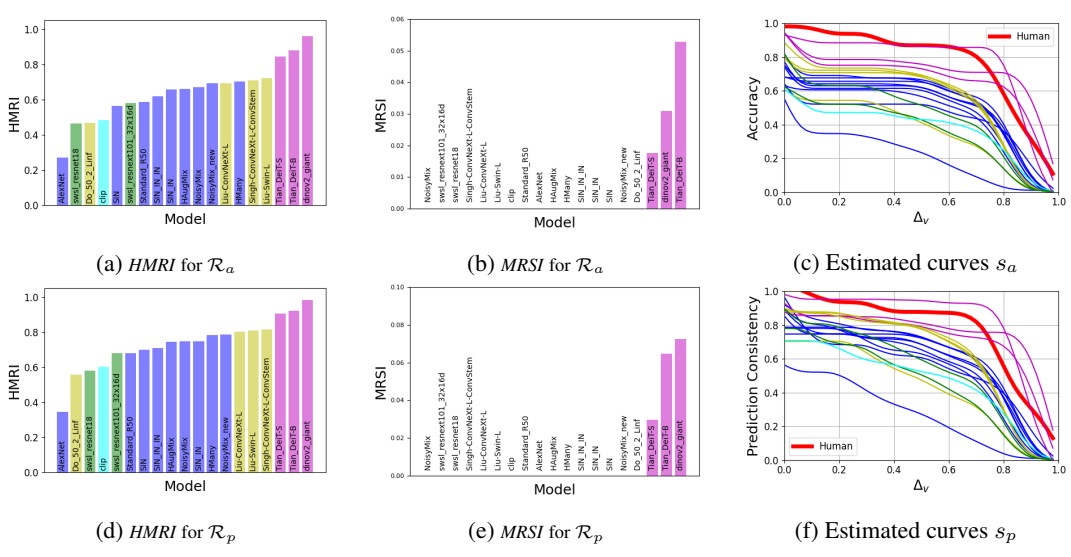

Figure 31: VCR evaluation results for Color Jitter.

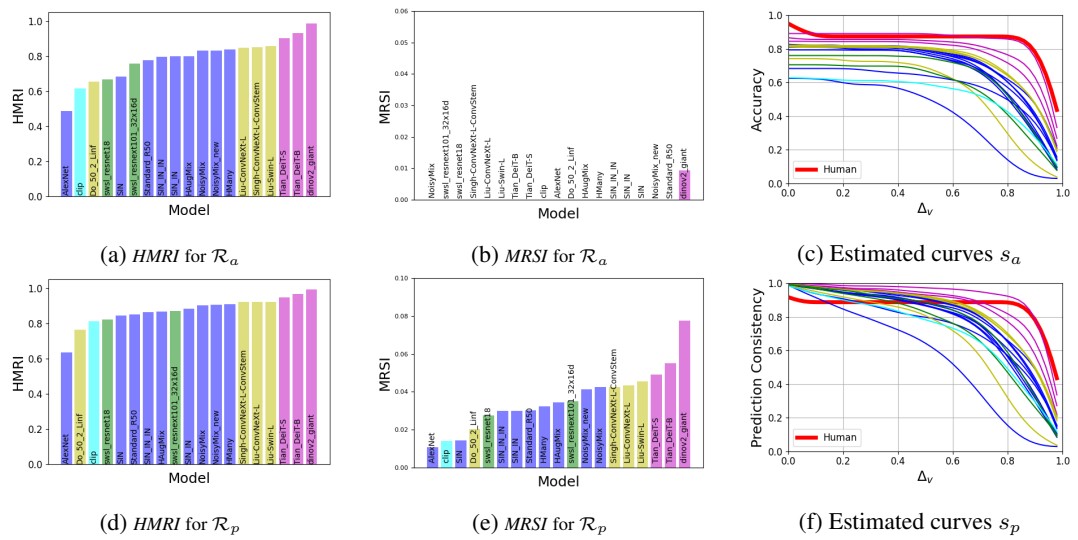

Figure 32: VCR evaluation results for Brightness.

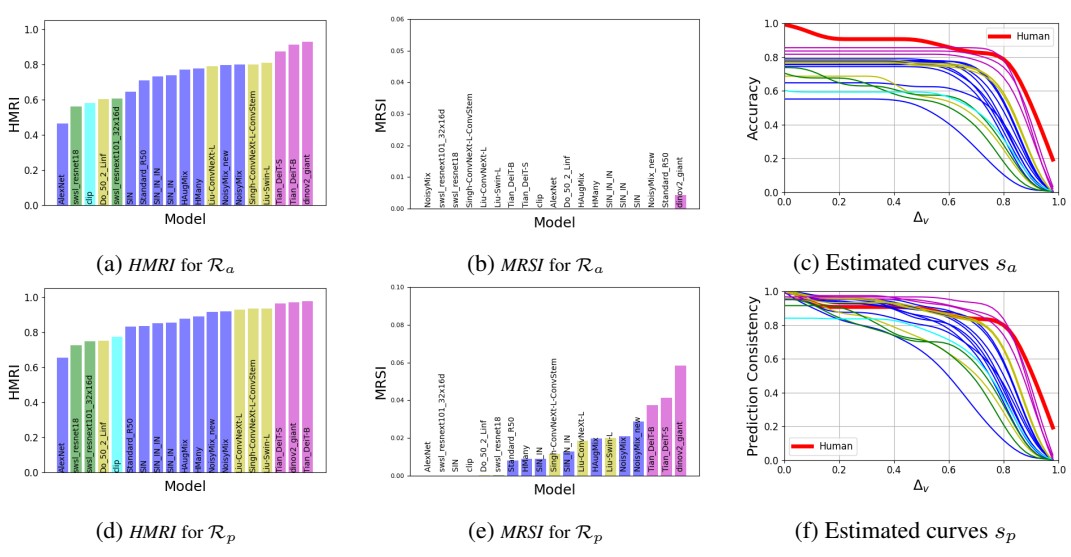

Figure 33: VCR evaluation results for Frost.