# OpenReview forum: "Assessing Visually-Continuous Corruption Robustness of Neural Networks Relative to Human Performance"
_ICLR.cc/2024/Conference — Submitted to ICLR 2024_

### Official Review · Reviewer_wHpz · 2023-10-31

**Soundness:** 3 good
**Presentation:** 1 poor
**Contribution:** 3 good
**Rating:** 5
**Confidence:** 3

**Summary:**

The paper proposes a visually-continuous corruption robustness (VCR) metric based on the visual information fidelity (VIF) metric. Furthermore, the authors propose two human-aware metrics HMRI and MRSI. The key message is that the gap between neural network robustness and human robustness is larger than expected. Authors have conducted experiments with 14 different image corruption techniques with 7718 human participants and different SOTA neural networks models.

**Strengths:**

- The paper is well-motivated and the experiments are very extensive
- The problem discussed is important and interesting
- Implementation and data was made available by the authors

**Weaknesses:**

- The paper is very hard to read and necessary background is not introduced. For example, I would have liked an explanation what visual information fidelity is. Overall, there is a lot of content squeezed in the 9 pages which makes the paper mostly incomprehensible.
- Due to above issue, I strongly suggest to publish the paper in a journal (which usually have no page limits). The quality of the write-up would highly benefit from this.
- Section 2 (Methods) needs a major rewrite to make it more accessible to readers not familiar with image quality metrics. Here are some points that can be improved:
    1. The section mentions multiple times that $\Delta_v \in [0,1]$.
    2. The variable c is used before it was defined.
    3. Authors should stress that $\Delta_v$ is just an auxiliary quantity that later is used to define the VCR.
    4. Authors could consider adding a figure to give an overview of the used and introduced metrics. As a reader, I was overwhelmed by all these acronyms. An overview would have been very helpful.
- I very much appreciate that the authors shared their code, however I find it inappropriate to refer to it as a "toolbox". In my opinion a toolbox is an installable Python package that is easily applicable to different models and datasets. Authors have to spend more time on their code repository before calling it a "toolbox".

Minor details:
- page 2: $max$ should be $\max$
- Tbl. 2 -> Tab. 2
- typo page 4: “coverages= of” and “[0..1]” (should be [0,1])
- Two paragraphs in the abstract look unusual
- Suggestions: "Uniform(0,1)" -> "U(0,1)"

**Questions:**

- What is impulse noise or glass blur?

**Details Of Ethics Concerns:**

The paper proposes a study with human participants (Mechanical Turk platform). Whether platforms like this are ethical or not is debatable. It is probably ok, but it would be good if someone familiar with the ICLR ethics guidelines would look into this.

---

> ### Author Response · Authors · 2023-11-22
> **Response to Reviewer wHpz**
>
> We appreciate the reviewer's comments on the difficulty to understand the text. We have revised Section 2 (Methods) as suggested, and replaced "toolbox" by "code".
>
> Most importantly, we have also created a visual summary of the VCR metrics in Section 8.4, and referred to the figures from Section 2.
>
> Further, we have added a more precise, but still high-level description of how VIF is computed, using wavelet decomposition and mutual information, in Section 8.5. The detailed formulas could easily fill an entire page but are available in the original paper [Sheikh & Bovik, 2006].
>
> Finally, regarding impulse noise or glass blur: impulse noise is a color analogue of salt-and-pepper noise, and glass blur simulates frosted glass windows or panels. Both corruptions are included in ImageNet-C. All of the corruptions in our study are visualized in Figure 10 in Section 8.6 of the appendix.

---

### Official Review · Reviewer_ZhQE · 2023-10-31

**Soundness:** 3 good
**Presentation:** 3 good
**Contribution:** 3 good
**Rating:** 6
**Confidence:** 4

**Summary:**

In this paper, the visually-continuous corruption robustness of existing Neural Networks (NNs) is examined and compared. In particular, two metrics including the Human-Relative Model Robustness Index (HMRI) and Model Robustness Superiority Index (MRSI) are proposed for the models’ performance evaluation on 14 corruption types. The experiments reveal the high robustness gap between humans and NNs.

**Strengths:**

1.	The proposed metrics are reasonable which could mitigate the evaluation bias caused by the quality distribution of the test set.
2.	Several interesting and meaningful observations are presented. For example, the performance of prediction consistency of different NNs is compared, in addition to the model accuracy.
3.	The authors explore the visually similar transformations, offering opportunities for more cost-effective robustness assessments.

**Weaknesses:**

1.	In this paper, The VIF is adopted as the quality measure. However, compared with the advanced full-reference quality measures, such as LPIPS [1], and DISTS [2], the VIF is usually inferior.
2.	In Sec.3, Page 4, the coverage between the IMAGENET-C and VCR Test Set is compared by splitting the full quality range into 40 bins. However, the coverage is highly relevant to the number of bins. In an extreme case, when the bin number is 1, the same coverage the two sets will possess. As such, how to ensure the coverage is reasonable?
3.	During subjective testing, human decisions are usually highly affected by the memory effect, i.e., a severally corrupted image could still be recognized successfully when humans have observed the same image content but with a high quality. The authors should illustrate how to avoid such effect and provide more details to demonstrate the reliability of the human decision collection.
4.	Typos: in Deﬁnition 1 [Human-Relative Model Robustness Index (HMRI)]: the S^m({\gamma}^(v)) should be S^m_{\gamma}^(v).

**Questions:**

Please see above.

---

> ### Author Response · Authors · 2023-11-22
> **Response to Reviewer ZhQE**
>
> We very much appreciate the valuable comments and questions. We address each of them individually.
>
> *"In this paper, The VIF is adopted as the quality measure. However, compared with the advanced full-reference quality measures, such as LPIPS [1], and DISTS [2], the VIF is usually inferior."*
>
> We choose VIF, since it is well-established, computationally efficient, applicable to our transformations, and still performing competitively compared to newer metrics. In fact, according to Table 1 in DISTS [Ding et al., 2022], VIF performs better than LPIPS across all three datasets used in the comparison, and it outperforms DISTS on one of them. Even though DNN-based metrics like DISTS may be applicable to a wider class of transformations than VIF, including those that affect both structure and textures, their scope may depend on the training datasets in potentially unpredictable ways. On the other hand, the scope of VIF is well-defined based on the metric's mathematical definition. In particular, VIF is suitable for evaluating corruption functions that can be locally described as a combination of signal attenuation and additive Gaussian noise in the sub-bands of the wavelet domain. The transformations in our experiments are local corruptions that are well within this scope.  We have updated Section 8.5 to reflect this rationale. However, future work should explore VCR using other IQA metrics.
>
>
> *"In Sec.3, Page 4, the coverage between the IMAGENET-C and VCR Test Set is compared by splitting the full quality range into 40 bins. However, the coverage is highly relevant to the number of bins. In an extreme case, when the bin number is 1, the same coverage the two sets will possess. As such, how to ensure the coverage is reasonable?"*
>
> We follow the established standard for the number of recall positions when calculating average precision, which are also empirical performance curves with a similar shape. For example, the KITTI benchmark has increased this number to 40 from 11, which was originally used in the PASCAL VOC benchmark.
>
> *"During subjective testing, human decisions are usually highly affected by the memory effect, i.e., a severely corrupted image could still be recognized successfully when humans have observed the same image content but with a high quality. The authors should illustrate how to avoid such effect and provide more details to demonstrate the reliability of the human decision collection."*
>
> The same original image, corrupted or not, was never shown again to the same participant, for exactly this reason. We’ve added this clarification to the experiment description.

---

### Official Review · Reviewer_gtM1 · 2023-11-01

**Soundness:** 3 good
**Presentation:** 3 good
**Contribution:** 3 good
**Rating:** 8
**Confidence:** 4

**Summary:**

Authors propose a new concept called visually-continuous corruption robustness (VCR), a better alternative to measure corruption robustness than ImageNet-C benchmark. Unlike pre-defined and definite parameters in ImageNet-C, this work creates a benchmark comprising continuous range of image corruption levels and evaluate human performance on the benchmark. Following that, two human aware metrics are introduced to compare neural network performance against humans. Authors demonstrate a notable disparity in robustness between the networks and human performance, despite the improvements seen in ImageNet-C benchmark.

**Strengths:**

-	Well written paper.

-	Thoughtful in designing the benchmark with visually-continuous corruptions.

-	Well detailed and carefully conducted human experiments.

-	Quantitatively shown that ImageNet-C comprise less coverage of visual corruptions than the proposed benchmark.

-	VCR is shown to be better robustness estimate than benchmarking on ImageNet-C -> Models having good performance on ImageNet-C shown to be not robust enough on the proposed benchmark.

-	This work emphasizes that model robustness is reliable upon verifying across continuous range of image corruption levels, instead of checking at pre-defined parameters.

-	Open-sourced with all human data. This benchmark is beneficial and steer the future research on corruption robustness in the right direction.

**Weaknesses:**

I don’t have major concerns about this work. I appreciate authors for considering wide range of models. However, some of the top performing robust models (ImageNet-C leaderboard https://paperswithcode.com/sota/domain-generalization-on-imagenet-c?p=augmix-a-simple-data-processing-method-to) like DINOv2, and MAE are missing in the evaluation. It is helpful to understand behaviour of these models in VCR.

**Questions:**

-	In page 3, under Testing VCR, it is mentioned that “only sufficient data in each group but not uniformity”. Why this is the case? How do you define the sufficient data here? What are the drawbacks of considering uniformity. It is mentioned that “ this specific design removes the possibility of biased results”, can you clarify what kind of biased distribution of data is referred here?

-	Humans are presented with one image at a time for 200 ms? Isn’t it too short to notice the image? Are the human participants in an average recognize objects in the image within that time? Would it be safe to assume that human participants do even better job when presented with an image upto 1s? It is mentioned that time was set to ensure fairness. Are the machines classify each image with the same 200 ms?

-	Please briefly discuss the qualification tests and sanity checks aimed to filter the participants.

-	A curious question, What is the total number of participants before the filtration process?

-	Are same images seen by each human participant?

-	Please connect (\Cref) the text “Fig 1” to the Figure 1. Similarly, for other figures and tables.

---

> ### Author Response · Authors · 2023-11-22
> **Response to Reviewer gtM1**
>
> We very much appreciate the valuable comments and questions. We address each of them individually.
>
> *"However, some of the top performing robust models (ImageNet-C leaderboard) like DINOv2, and MAE are missing in the evaluation. It is helpful to understand behaviour of these models in VCR."*
>
> We have added DINOv2, the top performer on the ImageNet-C leaderboard, to our evaluation. It also performs best in terms of VCR, along with the other two already-included transformers (TianDeiT-A and -B). The identified robustness gaps compared to humans remain though for all three model, especially for blur corruptions, so the paper conclusion remains unchanged. We’ve run into some problems interfacing the existing MAE models with ImageNet, but should have this resolved for the final version of the paper.
>
> *"In page 3, under Testing VCR, it is mentioned that “only sufficient data in each group but not uniformity”. Why this is the case? How do you define the sufficient data here? What are the drawbacks of considering uniformity."*
>
> Achieving uniformity in VCR requires inverting the mapping from VCR values to parameter values, which would require a costly optimization loop, which is sketched in Sec 8.5. However, since we are fitting a spline, the simple Alg. 1 in Sec 8.5 can tolerate the variation in bin sizes, even if the coverage of the bins is not 100% (see Table 1). This simple algorithm is similar to how average precision for object detection is computed, by first monotonically interpolating the existing datapoints before computing the area under the curve.
>
> *"It is mentioned that “ this specific design removes the possibility of biased results”, can you clarify what kind of biased distribution of data is referred here?"*
>
> ImageNet-C computes robustness by averaging performance over a set of corruption parameter levels, and it does not fit a continuous function before computing the average. As a result, and because of the non-linear (and varying across transformations) relationship between the corruption parameters and their visual effect (as measured by VQA), the ImageNet-C approach leads to big gaps in coverage of the visual change range, such as covering only low or only high visual change levels (see Figure 11). For example, ImageNet-C has no samples with Delta v below 0.5 for Gaussian noise; see Figure 11e in the appendix. As a result, the robustness error reported by ImageNet-C for Gaussian noise is biased towards the far end of the visual change range. Conversely, the robustness error reported by ImageNet-C for Gaussian blur does not consider robustness in the far end (Figure 11o) and is thus biased towards the low end.
>
> *"Humans are presented with one image at a time for 200 ms? Isn’t it too short to notice the image? Are the human participants in an average recognize objects in the image within that time? Would it be safe to assume that human participants do even better job when presented with an image upto 1s? It is mentioned that time was set to ensure fairness. Are the machines classify each image with the same 200 ms?''*
>
> We follow an established experimental protocol from Gherios et al. 2019a and Hu et al., 2022. The human visual system (HVS) achieves object recognition with feed-forward processing from a single glance within its early (ventral) stream anywhere between 100 to 200ms (see https://www.ncbi.nlm.nih.gov/pmc/articles/PMC3129241/). Above 200ms, recurrent processing starts to play a role, where evidence is integrated over multiple scans of the image (like in test time augmentations), and it may eventually involve functions of the pre-frontal cortex. In that sense, as you point out, human performance will increase with time and the increased recursive processing and the subsequent conscious analysis of the image content in the pre-frontal cortex. We limit the experiment to 200ms to compare the performance of feed-forward DNNs with the feed-forward processing in the brain, where the comparison is fair.  This clearly would not be the case when the human subjects start to involve the full spectrum of their brain functions.
>
> (continued in the next post)

---

> > ### Author Response · Authors · 2023-11-22
> > **Response to Reviewer gtM1 (continued)**
> >
> > *"Please briefly discuss the qualification tests and sanity checks aimed to filter the participants."*
> >
> > For qualification, we give each participant a small sample qualification with 20 images using the same set up as the actual experiment ahead of it. These images are hand-picked by us to be easy to classify (big, clear object in the center). The goal here is to make sure that the participants fully understand what they are supposed to do.
> >
> > Additionally, during the actual experiment we hid sanity check images, which are also easy images, and monitored the accuracy on those images. The participants are expected to correctly classify these images. The goal for sanity check images is to filter out participants who selected random classes (even though they might have passed the first qualification test).
> >
> > *"A curious question, What is the total number of participants before the filtration process?"*
> >
> > Unfortunately we did not record data for users who failed the qualifications, so we do not have a number for this.
> >
> > *“Are same images seen by each human participant?”*
> >
> > Participants see randomly sampled images from the generated test set, so some images may overlap over different participants; however, each participant will never see the same original image (corrupted or not) again (to eliminate memory effects).

---

### Official Review · Reviewer_zDY4 · 2023-11-03

**Soundness:** 2 fair
**Presentation:** 1 poor
**Contribution:** 2 fair
**Rating:** 3
**Confidence:** 4

**Summary:**

The authors propose a comparative study of the robustness of neural networks on visual changes (image corruptions) compared to humans, and perform a larg user-study. This work proposes two measures (HMRI and MRSI) to compare results.

**Strengths:**

+ address of the important problem of robustness to image corruptions
+ contribution of data: the work contributes with data relative to human-based (large number) assessment of image classification robustness, that might be used to further study the problem

**Weaknesses:**

_difficult to read_
The paper is hard-to-follow, and the argumentations or explanations are given in an overworded way. Objectives are not clear, and so possible insights that one should gain from reading this work. This makes difficult also to grasp the conceptual contributions or take-aways expected from the experimental analysis and results.
It looks also strange that a paper that proposes a dataset of corruptions applied to images does not show images of such corruptions and how they relate with existing benchmarks.

_poor insights_
The paper does not provide insights on how the results should be used: to design new models? train existing architectures differently? or other. WHile the user-study and experimental analysis is large, there is little to none instructive conclusions.

_relation with related work missing or weak_
No discussion or comparison with existing work and consider continuous corruptions, such as ImageNet-P and ImageNet-CCC, or other benchmark datasets such as ImageNet-Cbar or ImageNet-3DCC.

_choice of the models_
the choice of the tested models is not motivated, neither perspectives on the type of architecture, training data and strategies (e.g. supervised learning, self-supervised, using ImageNet21K or LAION or other datasets, CNN vs transformers) are given.

**Questions:**

How the augmentations proposed in this paper compare with the continuously changing corruptions of ImageNet-CCC? Why the focus is only wrt ImageNet-C?

How the models are chosen (criteria, comparative perspective, etc.)?

---

> ### Author Response · Authors · 2023-11-22
> **Response to Reviewer zDY4**
>
> We thank the reviewer for valuable comments. We address each weakness and question listed in the review individually.
>
> *“Objectives are not clear, and so possible insights that one should gain from reading this work. This makes difficult also to grasp the conceptual contributions or take-aways expected from the experimental analysis and results.”*
>
> The key objective is to improve the measurement of robustness and comparison to human performance. The key take-away is that robustness needs to be measured over the full and continuous range of visual change to avoid biased results. The use of a visual quality metric to quantify this range allows standardizing the reference over different transformations, since each transformation has different parameters and effects. We have emphasized the key insights in the introduction and conclusions (see the statements highlighted in boldface in the conclusion).
>
> *"It looks also strange that a paper that proposes a dataset of corruptions applied to images does not show images of such corruptions and how they relate with existing benchmarks."*
>
> The image corruptions are shown in Fig. 10 in the supplementary appendix. It takes a full page, so there is no space for it in the main paper. We have added a reference to the figure from the main body of the paper. The transformations themselves are from the existing benchmarks, 9 from ImageNet-C, four from Albumentations, and one from the previous study by Gheiros et al., and this is specified on p. 4 of the main paper.
>
> *"The paper does not provide insights on how the results should be used: to design new models? train existing architectures differently? or other. While the user-study and experimental analysis is large, there is little to none instructive conclusions."*
>
> The main insight is that failing to measure robustness over the full and continuous range, as done in existing benchmarks, leads to biased results that do not fully reflect the actual robustness compared to humans, as supported by the experiments. The key message is: What is not measured properly, cannot be improved. Designing new models or training techniques is not in the scope of this paper.
>
>  (continued in the next post)

---

> > ### Author Response · Authors · 2023-11-22
> > **Response to Reviewer zDY4 (continued)**
> >
> > *"No discussion or comparison with existing work and consider continuous corruptions, such as ImageNet-P and ImageNet-CCC, or other benchmark datasets such as ImageNet-Cbar or ImageNet-3DCC."*
> > Also:
> > *" How the augmentations proposed in this paper compare with the continuously changing corruptions of ImageNet-CCC? Why the focus is only wrt ImageNet-C?"*
> >
> > ImageNet-P (Hendrycks & Dietterich, 2019), ImageNet-Cbar (Mintun et al. 2021), and ImageNet-3DCC (Kar et al., 2022) have been discussed in the original submission in Section 8.2, i.e., the related work section in the supplementary appendix. Because of the limited space, the related work in the main body of the paper focused on work comparing the visual change robustness of NNs in comparison to humans, which is also the main focus of this paper.
> >
> > In summary, ImageNet-P, ImageNet-Cbar, and ImageNet-3DCC all follow the ImageNet-C design to apply only five pre-selected values per parameter, and this is why we focus on ImageNet-C. All these benchmarks suffer from limited coverage of corruption levels because of using five pre-selected values per parameter. This coverage is both sparse and does not reflect the non-linear relationship between parameters and visual change as measured by visual quality assessment (VQA) metrics. (BTW, our paper does not propose any new corruptions or augmentations, but new robustness metrics and human performance data.)
> >
> > ImageNet-CCC is indeed the only prior work targeting a more continuous range of corruptions, by using 20 pre-selected values per parameter. It does not check the coverage in terms of the effects on the images, however, which we do using a VQA metric (VIF). Further, this work focuses on continuous changes over time for benchmarking test-time adaptation, which is very different from general robustness benchmarking. Finally, the dataset has not been released as of writing, the code that has been released 6 months ago but does not run out-of-the box, and the paper has not been published yet. Nevertheless we have added a brief discussion of ImageNet-CCC to our related work (Section 8.2) and cited the preprint.
> >
> > In contrast to all these previous works, our method randomly and uniformly samples parameter values to cover the full range of visual change that a corruption function can achieve, which is modeled and assessed for coverage using a VQA metric. Defining the metrics metrics over the VQA range allows direct comparison of robustness values across transformations. This is in contrast to the previous works, which measure robustness by averaging performance over transformation parameter value ranges. Finally, our work also compares robustness of NNs with humans.
> >
> > *"The choice of the tested models is not motivated, neither perspectives on the type of architecture, training data and strategies (e.g. supervised learning, self-supervised, using ImageNet21K or LAION or other datasets, CNN vs transformers) are given."*
> > Also: *"How the models are chosen (criteria, comparative perspective, etc.)?"*
> >
> > We have added an overview of the main characteristics of the models (see Table 8.8 in the appendix). The models span different architectures (different CNNs and transformers) and training methods (supervised, adversarial, semi-weakly, and self-supervised).
> >
> > *"Difficult to read"*
> >
> > We have added a visual summary of the proposed metrics in Section 8.4 and made another editing pass over the main paper text to improve readability.

---

### Meta-Review · Area_Chair_LMfh · 2023-12-11

**Metareview:**

This work compares neural networks with humans against visually-continuous corruptions. Following that, two human-aware metrics are introduced to compare neural network performance against humans. The authors demonstrate a notable disparity in robustness between the networks and human performance.

While the authors have addressed some of the reviewers' concerns, some still remain. For example, it is less impactful to only focus on image corruption to evaluate robustness gaps between models and humans, as the robustness difference between them can be in many different ways. Consequently,  the detailed analysis of diverse image corruption types is limited in scope. In addition, the insights derived from this work are limited as no deep understanding is developed to explain why this robustness gap stands and no further methods are derived from the evaluation to enhance robustness.

**Justification For Why Not Higher Score:**

Based on the limited research scope and impacts on the research community, we recommend rejecting this work.

**Justification For Why Not Lower Score:**

N/A

---

### Decision · Program_Chairs · 2024-01-16

Reject